# Accelerated Proximal Gradient Methods for Nonconvex Programming

**Huan Li**         **Zhouchen Lin** ✉
Key Lab. of Machine Perception (MOE), School of EECS, Peking University, P. R. China
Cooperative Medianet Innovation Center, Shanghai Jiaotong University, P. R. China
`lihuanss@pku.edu.cn`        `zlin@pku.edu.cn`

## Abstract

Nonconvex and nonsmooth problems have recently received considerable attention in signal/image processing, statistics and machine learning. However, solving the nonconvex and nonsmooth optimization problems remains a big challenge. Accelerated proximal gradient (APG) is an excellent method for convex programming. However, it is still unknown whether the usual APG can ensure the convergence to a critical point in nonconvex programming. In this paper, we extend APG for general nonconvex and nonsmooth programs by introducing a monitor that satisfies the sufficient descent property. Accordingly, we propose a monotone APG and a nonmonotone APG. The latter waives the requirement on monotonic reduction of the objective function and needs less computation in each iteration. To the best of our knowledge, we are the first to provide APG-type algorithms for general nonconvex and nonsmooth problems ensuring that every accumulation point is a critical point, and the convergence rates remain $O\left(\frac{1}{k^2}\right)$ when the problems are convex, in which $k$ is the number of iterations. Numerical results testify to the advantage of our algorithms in speed.

## 1 Introduction

In recent years, sparse and low rank learning has been a hot research topic and leads to a wide variety of applications in signal/image processing, statistics and machine learning. $l_1$-norm and nuclear norm, as the continuous and convex surrogates of $l_0$-norm and rank, respectively, have been used extensively in the literature. See e.g., the recent collections [1]. Although $l_1$-norm and nuclear norm have achieved great success, in many cases they are suboptimal as they can promote sparsity and low-rankness only under very limited conditions [2, 3]. To address this issue, many nonconvex regularizers have been proposed, such as $l_p$-norm [4], Capped-$l_1$ penalty [3], Log-Sum Penalty [2], Minimax Concave Penalty [5], Geman Penalty [6], Smoothly Clipped Absolute Deviation [7] and Schatten-$p$ norm [8]. This trend motivates a revived interest in the analysis and design of algorithms for solving nonconvex and nonsmooth problems, which can be formulated as

$$\min_{\mathbf{x} \in \mathbb{R}^n} F(\mathbf{x}) = f(\mathbf{x}) + g(\mathbf{x}), \tag{1}$$

where $f$ is differentiable (it can be nonconvex) and $g$ can be both nonconvex and nonsmooth.

Accelerated gradient methods have been at the heart of convex optimization research. In a series of celebrated works [9, 10, 11, 12, 13, 14], several accelerated gradient methods are proposed for problem (1) with convex $f$ and $g$. In these methods, $k$ iterations are sufficient to find a solution within $O\left(\frac{1}{k^2}\right)$ error from the optimal objective value. Recently, Ghadimi and Lan [15] presented a unified treatment of accelerated gradient method (UAG) for convex, nonconvex and stochastic optimiza-

Table 1: Comparisons of GD (General Descent Method), iPiano, GIST, GDPA, IR, IFB, APG, UAG and our method for problem (1). The measurements include the assumption, whether the methods accelerate for convex programs (CP) and converge for nonconvex programs (NCP).

| Method name | Assumption | Accelerate (CP) | converge (NCP) |
|---|---|---|---|
| GD [16, 17] | $f + g$: KL | No | Yes |
| iPiano [18] | nonconvex $f$, convex $g$ | No | Yes |
| GIST [19] | nonconvex $f$, $g = g_1 - g_2$, $g_1, g_2$ convex | No | Yes |
| GDPA [20] | nonconvex $f$, $g = g_1 - g_2$, $g_1, g_2$ convex | No | Yes |
| IR [8, 21] | special $f$ and $g$ | No | Yes |
| IFB [22] | nonconvex $f$, nonconvex $g$ | No | Yes |
| APG [12, 13] | convex $f$, convex $g$ | Yes | Unclear |
| UAG [15] | nonconvex $f$, convex $g$ | Yes | Yes |
| Ours | nonconvex $f$, nonconvex $g$ | Yes | Yes |

tion. They proved that their algorithm converges[1] in nonconvex programming with nonconvex $f$ but convex $g$ and accelerates with an $O\left(\frac{1}{k^2}\right)$ convergence rate in convex programming for problem (1). Convergence rate about the gradient mapping is also analyzed in [15].

Attouch et al. [16] proposed a unified framework to prove the convergence of a general class of descent methods using the Kurdyka-Łojasiewicz (KL) inequality for problem (1) and Frankel et al. [17] studied the convergence rates of general descent methods under the assumption that the desingularising function $\varphi$ in KL property has the form of $\frac{C}{\theta}t^{\theta}$. A typical example in their framework is the proximal gradient method. However, there is no literature showing that there exists an accelerated gradient method satisfying the conditions in their framework.

Other typical methods for problem (1) includes Inertial Forward-Backward (IFB) [22], iPiano [18], General Iterative Shrinkage and Thresholding (GIST) [19], Gradient Descent with Proximal Average(GDPA) [20] and Iteratively Reweighted Algorithms (IR) [8, 21]. Table 1 demonstrates that the existing methods are not ideal. GD and IFB cannot accelerate the convergence for convex programs. GIST and GDPA require that $g$ should be explicitly written as a difference of two convex functions. iPiano demands the convexity of $g$ and IR is suitable for some special cases of problem (1). APG can accelerate the convergence for convex programs, however, it is unclear whether APG can converge to critical points for nonconvex programs. UAG can ensure the convergence for nonconvex programming, however, it requires $g$ to be convex. This restricts the applications of UAG to solving nonconvexly regularized problems, such as sparse and low rank learning. To the best of our knowledge, extending the accelerated gradient method for general nonconvex and nonsmooth programs while keeping the $O\left(\frac{1}{k^2}\right)$ convergence rate in the convex case remains an open problem.

In this paper we aim to extend Beck and Teboulle's APG [12, 13] to solve general nonconvex and nonsmooth problem (1). APG first extrapolates a point $\mathbf{y}_k$ by combining the current point and the previous point, then solves a proximal mapping problem. When extending APG to nonconvex programs the chief difficulty lies in the extrapolated point $\mathbf{y}_k$. We have little restriction on $F(\mathbf{y}_k)$ when the convexity is absent. In fact, $F(\mathbf{y}_k)$ can be arbitrarily larger than $F(\mathbf{x}_k)$ when $\mathbf{y}_k$ is a bad extrapolation, especially when $F$ is oscillatory. When $\mathbf{x}_{k+1}$ is computed by a proximal mapping at a bad $\mathbf{y}_k$, $F(\mathbf{x}_{k+1})$ may also be arbitrarily larger than $F(\mathbf{x}_k)$. Beck and Teboulle's monotone APG [12] ensures $F(\mathbf{x}_{k+1}) \leq F(\mathbf{x}_k)$. However, this is not enough to ensure the convergence to critical points. To address this issue, we introduce a monitor satisfying the sufficient descent property to prevent a bad extrapolation of $\mathbf{y}_k$ and then correct it by this monitor. In summary, our contributions include:

1. We propose APG-type algorithms for general nonconvex and nonsmooth programs (1). We first extend Beck and Teboulle's monotone APG [12] by replacing their *descent* condition with *sufficient descent* condition. This critical change ensures that every accumulation point is a critical point. Our monotone APG satisfies some *modified* conditions for the framework of [16, 17] and thus stronger results on convergence rate can be obtained under the KL

assumption. Then we propose a nonmonotone APG, which allows for larger stepsizes when line search is used and reduces the average number of proximal mappings in each iteration. Thus it can further speed up the convergence in practice.

2. For our APGs, the convergence rates maintain $O\left(\frac{1}{k^2}\right)$ when the problems are convex. This result is of great significance when the objective function is locally convex in the neighborhoods of local minimizers even if it is globally nonconvex.

## 2 Preliminaries

### 2.1 Basic Assumptions

Note that a function $g : \mathbb{R}^n \to (-\infty, +\infty]$ is said to be proper if dom $g \neq \emptyset$, where dom $g = \{\mathbf{x} \in \mathbb{R} : g(\mathbf{x}) < +\infty\}$. $g$ is lower semicontinuous at point $\mathbf{x}_0$ if $\liminf_{x \to \mathbf{x}_0} g(\mathbf{x}) \geq g(\mathbf{x}_0)$. In problem (1), we assume that $f$ is a proper function with Lipschitz continuous gradients and $g$ is proper and lower semicontinuous. We assume that $F(\mathbf{x})$ is coercive, i.e., $F$ is bounded from below and $F(\mathbf{x}) \to \infty$ when $\|\mathbf{x}\| \to \infty$, where $\|\cdot\|$ is the $l_2$-norm.

### 2.2 KL Inequality

**Definition 1.** *[23] A function $f : \mathbb{R}^n \to (-\infty, +\infty]$ is said to have the KL property at $\overline{\mathbf{u}} \in dom\partial f := \{\mathbf{x} \in \mathbb{R}^n : \partial f(\mathbf{u}) \neq \emptyset\}$ if there exists $\eta \in (0, +\infty]$, a neighborhood $U$ of $\overline{\mathbf{u}}$ and a function $\varphi \in \Phi_\eta$, such that for all $\mathbf{u} \in U \bigcap \{\mathbf{u} \in \mathbb{R}^n : f(\overline{\mathbf{u}}) < f(\mathbf{u}) < f(\overline{\mathbf{u}}) + \eta\}$, the following inequality holds*

$$\varphi'(f(\mathbf{u}) - f(\overline{\mathbf{u}}))dist(0, \partial f(\mathbf{u})) > 1, \tag{2}$$

*where $\Phi_\eta$ stands for a class of function $\varphi : [0, \eta) \to \mathbb{R}^+$ satisfying: (1) $\varphi$ is concave and $C^1$ on $(0, \eta)$; (2) $\varphi$ is continuous at 0, $\varphi(0) = 0$; and (3) $\varphi'(\mathbf{x}) > 0, \forall \mathbf{x} \in (0, \eta)$.*

All semi-algebraic functions and subanalytic functions satisfy the KL property. Specially, the desingularising function $\varphi(t)$ of semi-algebraic functions can be chosen to be the form of $\frac{C}{\theta}t^\theta$ with $\theta \in (0, 1]$. Typical semi-algebraic functions include real polynomial functions, $\|x\|_p$ with $p \geq 0$, rank$(X)$, the indicator function of PSD cone, Stiefel manifolds and constant rank matrices [23].

### 2.3 Review of APG in the Convex Case

We first review APG in the convex case. Bech and Teboulle [13] extend Nesterov's accelerated gradient method to the nonsmooth case. It is named the Accelerated Proximal Gradient method and consists of the following steps:

$$\mathbf{y}_k = \mathbf{x}_k + \frac{t_{k-1} - 1}{t_k}(\mathbf{x}_k - \mathbf{x}_{k-1}), \tag{3}$$

$$\mathbf{x}_{k+1} = \text{prox}_{\alpha_k g}(\mathbf{y}_k - \alpha_k \nabla f(\mathbf{y}_k)), \tag{4}$$

$$t_{k+1} = \frac{\sqrt{4(t_k)^2 + 1} + 1}{2}, \tag{5}$$

where the proximal mapping is defined as $\text{prox}_{\alpha g}(\mathbf{x}) = \text{argmin}_{\mathbf{u}} g(\mathbf{u}) + \frac{1}{2\alpha}\|\mathbf{x} - \mathbf{u}\|^2$. APG is not a monotone algorithm, which means that $F(\mathbf{x}_{k+1})$ may not be smaller than $F(\mathbf{x}_k)$. So Beck and Teboulle [12] further proposed a monotone APG, which consists of the following steps:

$$\mathbf{y}_k = \mathbf{x}_k + \frac{t_{k-1}}{t_k}(\mathbf{z}_k - \mathbf{x}_k) + \frac{t_{k-1} - 1}{t_k}(\mathbf{x}_k - \mathbf{x}_{k-1}), \tag{6}$$

$$\mathbf{z}_{k+1} = \text{prox}_{\alpha_k g}(\mathbf{y}_k - \alpha_k \nabla f(\mathbf{y}_k)), \tag{7}$$

$$t_{k+1} = \frac{\sqrt{4(t_k)^2 + 1} + 1}{2}, \tag{8}$$

$$\mathbf{x}_{k+1} = \begin{cases} \mathbf{z}_{k+1}, & \text{if } F(\mathbf{z}_{k+1}) \leq F(\mathbf{x}_k), \\ \mathbf{x}_k, & \text{otherwise.} \end{cases} \tag{9}$$

# 3 APGs for Nonconvex Programs

In this section, we propose two APG-type algorithms for general nonconvex nonsmooth problems. We establish the convergence in the nonconvex case and the $O\left(\frac{1}{k^2}\right)$ convergence rate in the convex case. When the KL property is satisfied we also provide stronger results on convergence rate.

## 3.1 Monotone APG

We give two reasons that result in the difficulty of convergence analysis on the usual APG [12, 13] for nonconvex programs: (1) $\mathbf{y}_k$ may be a bad extrapolation, (2) in [12] only descent property, $F(\mathbf{x}_{k+1}) \leq F(\mathbf{x}_k)$, is ensured. To address these issues, we need to monitor and correct $\mathbf{y}_k$ when it has the potential to fail, and the monitor should enjoy the property of *sufficient descent* which is critical to ensure the convergence to a critical point. As is known, proximal gradient methods can make sure sufficient descent [16] (cf. (15)). So we use a proximal gradient step as the monitor. More specially, our algorithm consists of the following steps:

$$\mathbf{y}_k = \mathbf{x}_k + \frac{t_{k-1}}{t_k}(\mathbf{z}_k - \mathbf{x}_k) + \frac{t_{k-1}-1}{t_k}(\mathbf{x}_k - \mathbf{x}_{k-1}), \tag{10}$$

$$\mathbf{z}_{k+1} = \text{prox}_{\alpha_y g}(\mathbf{y}_k - \alpha_y \nabla f(\mathbf{y}_k)), \tag{11}$$

$$\mathbf{v}_{k+1} = \text{prox}_{\alpha_x g}(\mathbf{x}_k - \alpha_x \nabla f(\mathbf{x}_k)), \tag{12}$$

$$t_{k+1} = \frac{\sqrt{4(t_k)^2 + 1} + 1}{2}, \tag{13}$$

$$\mathbf{x}_{k+1} = \begin{cases} \mathbf{z}_{k+1}, & \text{if } F(\mathbf{z}_{k+1}) \leq F(\mathbf{v}_{k+1}), \\ \mathbf{v}_{k+1}, & \text{otherwise.} \end{cases} \tag{14}$$

where $\alpha_y$ and $\alpha_x$ can be fixed constants satisfying $\alpha_y < \frac{1}{L}$ and $\alpha_x < \frac{1}{L}$, or dynamically computed by backtracking line search initialized by Barzilai-Borwein rule[2]. $L$ is the Lipschitz constant of $\nabla f$.

Our algorithm is an extension of Beck and Teboulle's monotone APG [12]. The difference lies in the extra $\mathbf{v}$, as the role of monitor, and the correction step of $\mathbf{x}$-update. In (9) $F(\mathbf{z}_{k+1})$ is compared with $F(\mathbf{x}_k)$, while in (14) $F(\mathbf{z}_{k+1})$ is compared with $F(\mathbf{v}_{k+1})$. A further difference is that Beck and Teboulle's algorithm only ensures descent while our algorithm makes sure sufficient descent, which means

$$F(\mathbf{x}_{k+1}) \leq F(\mathbf{x}_k) - \delta\|\mathbf{v}_{k+1} - \mathbf{x}_k\|^2, \tag{15}$$

where $\delta > 0$ is a small constant. It is not difficult to understand that only the descent property cannot ensure the convergence to a critical point in nonconvex programming. We present our convergence result in the following theorem[3].

**Theorem 1.** *Let $f$ be a proper function with Lipschitz continuous gradients and $g$ be proper and lower semicontinuous. For nonconvex $f$ and nonconvex nonsmooth $g$, assume that $F(x)$ is coercive. Then $\{\mathbf{x}_k\}$ and $\{\mathbf{v}_k\}$ generated by (10)-(14) are bounded. Let $\mathbf{x}^*$ be any accumulation point of $\{\mathbf{x}_k\}$, we have $0 \in \partial F(\mathbf{x}^*)$, i.e., $\mathbf{x}^*$ is a critical point.*

A remarkable aspect of our algorithm is that although we have made some modifications on Beck and Teboulle's algorithm, the $O\left(\frac{1}{k^2}\right)$ convergence rate in the convex case still holds. Similar to Theorem 5.1 in [12], we have the following theorem on the accelerated convergence in the convex case:

**Theorem 2.** *For convex $f$ and $g$, assume that $\nabla f$ is Lipschitz continuous, let $\mathbf{x}^*$ be any global optimum, then $\{\mathbf{x}_k\}$ generated by (10)-(14) satisfies*

$$F(\mathbf{x}_{N+1}) - F(\mathbf{x}^*) \leq \frac{2}{\alpha_y(N+1)^2}\|\mathbf{x}_0 - \mathbf{x}^*\|^2, \tag{16}$$

When the objective function is locally convex in the neighborhood of local minimizers, Theorem 2 means that APG can ensure to have an $O\left(\frac{1}{k^2}\right)$ convergence rate when approaching to a local minimizer, thus accelerating the convergence.

For better reference, we summarize the proposed monotone APG algorithm in Algorithm 1.

**Algorithm 1** Monotone APG

---
Initialize $\mathbf{z}_1 = \mathbf{x}_1 = \mathbf{x}_0$, $t_1 = 1$, $t_0 = 0$, $\alpha_y < \frac{1}{L}$, $\alpha_x < \frac{1}{L}$.
**for** $k = 1, 2, 3, \cdots$ **do**
    update $\mathbf{y}_k$, $\mathbf{z}_{k+1}$, $\mathbf{v}_{k+1}$, $t_{k+1}$ and $\mathbf{x}_{k+1}$ by (10)-(14).
**end for**

---

## 3.2 Convergence Rate under the KL Assumption

The KL property is a powerful tool and is studied by [16], [17] and [23] for a class of general descent methods. The usual APG in [12, 13] does not satisfy the sufficient descent property, which is crucial to use the KL property, and thus has no conclusions under the KL assumption. On the other hand, due to the intermediate variables $\mathbf{y}_k$, $\mathbf{v}_k$ and $\mathbf{z}_k$, our algorithm is more complex than the general descent methods and also does *not* satisfy the conditions therein. However, due to the monitor-corrector step (12) and (14), some *modified* conditions[4] can be satisfied and we can still get some exciting results under the KL assumption. With the same framework of [17], we have the following theorem.

**Theorem 3.** *Let $f$ be a proper function with Lipschitz continuous gradients and $g$ be proper and lower semicontinuous. For nonconvex $f$ and nonconvex nonsmooth $g$, assume that $F(x)$ is coercive. If we further assume that $f$ and $g$ satisfy the KL property and the desingularising function has the form of $\varphi(t) = \frac{C}{\theta} t^\theta$ for some $C > 0$, $\theta \in (0, 1]$, then*

1. *If $\theta = 1$, then there exists $k_1$ such that $F(\mathbf{x}_k) = F^*$ for all $k > k_1$ and the algorithm terminates in finite steps.*

2. *If $\theta \in [\frac{1}{2}, 1)$, then there exists $k_2$ such that for all $k > k_2$,*

$$F(\mathbf{x}_k) - F^* \leq \left( \frac{d_1 C^2}{1 + d_1 C^2} \right)^{k - k_2} r_{k_2}. \tag{17}$$

3. *If $\theta \in (0, \frac{1}{2})$, then there exists $k_3$ such that for all $k > k_3$,*

$$F(\mathbf{x}_k) - F^* \leq \left( \frac{C}{(k - k_3) d_2 (1 - 2\theta)} \right)^{\frac{1}{1 - 2\theta}}, \tag{18}$$

*where $F^*$ is the same function value at all the accumulation points of $\{\mathbf{x}_k\}$, $r_k = F(\mathbf{v}_k) - F^*$, $d_1 = \left( \frac{1}{\alpha_x} + L \right)^2 / \left( \frac{1}{2\alpha_x} - \frac{L}{2} \right)$ and $d_2 = \min \left\{ \frac{1}{2d_1 C}, \frac{C}{1 - 2\theta} \left( 2^{\frac{2\theta - 1}{2\theta - 2}} - 1 \right) r_0^{2\theta - 1} \right\}$*

When $F(\mathbf{x})$ is a semi-algebraic function, the desingularising function $\varphi(t)$ can be chosen to be the form of $\frac{C}{\theta} t^\theta$ with $\theta \in (0, 1]$ [23]. In this case, as shown in Theorem 3, our algorithm converges in finite iterations when $\theta = 1$, converges with a linear rate when $\theta \in [\frac{1}{2}, 1)$ and a sublinear rate (at least $O(\frac{1}{k})$) when $\theta \in (0, \frac{1}{2})$ for the gap $F(\mathbf{x}_k) - F^*$. This is the same as the results mentioned in [17], although our algorithm does *not* satisfy the conditions therein.

## 3.3 Nonmonotone APG

Algorithm 1 is a monotone algorithm. When the problem is ill-conditioned, a monotone algorithm has to creep along the bottom of a narrow curved valley so that the objective function value does not increase, resulting in short stepsizes or even zigzagging and hence slow convergence [24]. Removing the requirement on monotonicity can improve convergence speed because larger stepsizes can be adopted when line search is used.

On the other hand, in Algorithm 1 we need to compute $\mathbf{z}_{k+1}$ and $\mathbf{v}_{k+1}$ in each iteration and use $\mathbf{v}_{k+1}$ to monitor and correct $\mathbf{z}_{k+1}$. This is a conservative strategy. In fact, we can accept $\mathbf{z}_{k+1}$ as $\mathbf{x}_{k+1}$ directly if it satisfies some criterion showing that $\mathbf{y}_k$ is a good extrapolation. Then $\mathbf{v}_{k+1}$ is computed only when this criterion is not met. Thus, we can reduce the average number of proximal

mappings, accordingly the computation cost, in each iteration. So in this subsection we propose a nonmonotone APG to speed up convergence.

In monotone APG, (15) is ensured. In nonmonotone APG, we allow $\mathbf{x}_{k+1}$ to make a larger objective function value than $F(\mathbf{x}_k)$. Specifically, we allow $\mathbf{x}_{k+1}$ to yield an objective function value smaller than $c_k$, a relaxation of $F(\mathbf{x}_k)$. $c_k$ should not be too far from $F(\mathbf{x}_k)$. So the average of $F(\mathbf{x}_k), F(\mathbf{x}_{k-1}), \cdots, F(\mathbf{x}_1)$ is a good choice. Thus we follow [24] to define $c_k$ as a convex combination of $F(\mathbf{x}_k), F(\mathbf{x}_{k-1}), \cdots, F(\mathbf{x}_1)$ with exponentially decreasing weights:

$$c_k = \frac{\sum_{j=1}^k \eta^{k-j} F(\mathbf{x}_j)}{\sum_{j=1}^k \eta^{k-j}}, \tag{19}$$

where $\eta \in [0, 1)$ controls the degree of nonmonotonicity. In practice $c_k$ can be efficiently computed by the following recursion:

$$q_{k+1} = \eta q_k + 1, \tag{20}$$

$$c_{k+1} = \frac{\eta q_k c_k + F(\mathbf{x}_{k+1})}{q_{k+1}}, \tag{21}$$

where $q_1 = 1$ and $c_1 = F(\mathbf{x}_1)$.

According to (14), we can split (15) into two parts by the different choices of $\mathbf{x}_{k+1}$. Accordingly, in nonmonotone APG we consider the following two conditions to replace (15):

$$F(\mathbf{z}_{k+1}) \leq c_k - \delta\|\mathbf{z}_{k+1} - \mathbf{y}_k\|^2, \tag{22}$$

$$F(\mathbf{v}_{k+1}) \leq c_k - \delta\|\mathbf{v}_{k+1} - \mathbf{x}_k\|^2. \tag{23}$$

We choose (22) as the criteria mentioned before. When (22) holds, we deem that $\mathbf{y}_k$ is a good extrapolation and accept $\mathbf{z}_{k+1}$ directly. Then we do not compute $\mathbf{v}_{k+1}$ in this case. However, (22) does not hold all the time. When it fails, we deem that $\mathbf{y}_k$ may not be a good extrapolation. In this case, we compute $\mathbf{v}_{k+1}$ by (12) satisfying (23), and then monitor and correct $\mathbf{z}_{k+1}$ by (14). (23) is ensured when $\alpha_x \leq 1/L$. When backtracking line search is used, such $\mathbf{v}_{k+1}$ that satisfies (23) can be found in finite steps[5].

Combing (20), (21), (22) and $\mathbf{x}_{k+1} = \mathbf{z}_{k+1}$ we have

$$c_{k+1} \leq c_k - \frac{\delta\|\mathbf{x}_{k+1} - \mathbf{y}_k\|^2}{q_{k+1}}. \tag{24}$$

Similarly, replacing (22) and $\mathbf{x}_{k+1} = \mathbf{z}_{k+1}$ by (23) and $\mathbf{x}_{k+1} = \mathbf{v}_{k+1}$, respectively, we have

$$c_{k+1} \leq c_k - \frac{\delta\|\mathbf{x}_{k+1} - \mathbf{x}_k\|^2}{q_{k+1}}. \tag{25}$$

This means that we replace the sufficient descent condition of $F(\mathbf{x}_k)$ in (15) by the sufficient descent of $c_k$.

We summarize the nonmonotone APG in Algorithm 2[6]. Similar to monotone APG, nonmonotone APG also enjoys the convergence property in the nonconvex case and the $O\left(\frac{1}{k^2}\right)$ convergence rate in the convex case. We present our convergence result in Theorem 4. Theorem 2 still holds for Algorithm 2 with no modification. So we omit it here.

Define $\Omega_1 = \{k_1, k_2, \cdots, k_j, \cdots\}$ and $\Omega_2 = \{m_1, m_2, \cdots, m_j, \cdots\}$, such that in Algorithm 2, (22) holds and $\mathbf{x}_{k+1} = \mathbf{z}_{k+1}$ is executed for all $k = k_j \in \Omega_1$. For all $k = m_j \in \Omega_2$, (22) does not hold and (14) is executed. Then we have $\Omega_1 \bigcap \Omega_2 = \emptyset$, $\Omega_1 \bigcup \Omega_2 = \{1, 2, 3, \cdots, \}$ and the following theorem holds.

**Theorem 4.** *Let $f$ be a proper function with Lipschitz continuous gradients and $g$ be proper and lower semicontinuous. For nonconvex $f$ and nonconvex nonsmooth $g$, assume that $F(x)$ is coercive. Then $\{\mathbf{x}_k\}$, $\{\mathbf{v}_k\}$ and $\{\mathbf{y}_{k_j}\}$ where $k_j \in \Omega_1$ generated by Algorithm 2 are bounded, and*

    *1. if $\Omega_1$ or $\Omega_2$ is finite, then for any accumulation point $\{\mathbf{x}^*\}$ of $\{\mathbf{x}_k\}$, we have $0 \in \partial F(\mathbf{x}^*)$.*

**Algorithm 2** Nonmonotone APG

---

Initialize $\mathbf{z}_1 = \mathbf{x}_1 = \mathbf{x}_0, t_1 = 1, t_0 = 0, \eta \in [0,1), \delta > 0, c_1 = F(\mathbf{x}_1), q_1 = 1, \alpha_x < \frac{1}{L}, \alpha_y < \frac{1}{L}$.

**for** $k = 1, 2, 3, \cdots$ **do**

    $\mathbf{y}_k = \mathbf{x}_k + \frac{t_{k-1}}{t_k}(\mathbf{z}_k - \mathbf{x}_k) + \frac{t_{k-1}-1}{t_k}(\mathbf{x}_k - \mathbf{x}_{k-1})$,

    $\mathbf{z}_{k+1} = \text{prox}_{\alpha_y g}(\mathbf{y}_k - \alpha_y \nabla f(\mathbf{y}_k))$

    **if** $F(\mathbf{z}_{k+1}) \leq c_k - \delta\|\mathbf{z}_{k+1} - \mathbf{y}_k\|^2$ **then**

        $\mathbf{x}_{k+1} = \mathbf{z}_{k+1}$.

    **else**

        $\mathbf{v}_{k+1} = \text{prox}_{\alpha_x g}(\mathbf{x}_k - \alpha_x \nabla f(\mathbf{x}_k))$,

        $\mathbf{x}_{k+1} = \begin{cases} \mathbf{z}_{k+1}, & \text{if } F(\mathbf{z}_{k+1}) \leq F(\mathbf{v}_{k+1}), \\ \mathbf{v}_{k+1}, & \text{otherwise.} \end{cases}$

    **end if**

    $t_{k+1} = \frac{\sqrt{4(t_k)^2+1}+1}{2}$,

    $q_{k+1} = \eta q_k + 1$,

    $c_{k+1} = \frac{\eta q_k c_k + F(\mathbf{x}_{k+1})}{q_{k+1}}$.

**end for**

---

    2. *if $\Omega_1$ and $\Omega_2$ are both infinite, then for any accumulation point $\mathbf{x}^*$ of $\{\mathbf{x}_{k_j+1}\}$, $\mathbf{y}^*$ of $\{\mathbf{y}_{k_j}\}$ where $k_j \in \Omega_1$ and any accumulation point $\mathbf{v}^*$ of $\{\mathbf{v}_{m_j+1}\}$, $\mathbf{x}^*$ of $\{\mathbf{x}_{m_j}\}$ where $m_j \in \Omega_2$, we have $0 \in \partial F(\mathbf{x}^*)$, $0 \in \partial F(\mathbf{y}^*)$ and $0 \in \partial F(\mathbf{v}^*)$.*

## 4 Numerical Results

In this section, we test the performance of our algorithm on the problem of Sparse Logistic Regression (LR)[7].Sparse LR is an attractive extension to LR as it can reduce overfitting and perform feature selection simultaneously. Sparse LR is widely used in areas such as bioinformatics [25] and text categorization [26]. In this subsection, we follow Gong et al. [19] to consider Sparse LR with a nonconvex regularizer:

$$\min_{\mathbf{w}} \frac{1}{n} \sum_{i=1}^{n} \log(1 + \exp(-y_i \mathbf{x}_i^T \mathbf{w})) + r(\mathbf{w}). \tag{26}$$

We choose $r(\mathbf{w})$ as the capped $l_1$ penalty [3], defined as

$$r(\mathbf{w}) = \lambda \sum_{i=1}^{d} \min(|w_i|, \theta), \quad \theta > 0. \tag{27}$$

We compare monotone APG (mAPG) and nonmonotone APG (nmAPG) with monotone GIST[8] (mGIST), nonmonotone GIST (nmGIST) [19] and IFB [22]. We test the performance on the real-sim data set[9], which contains 72309 samples of 20958 dimensions. We follow [19] to set $\lambda = 0.0001$, $\theta = 0.1\lambda$ and the starting point as zero vectors. In nmAPG we set $\eta = 0.8$. In IFB the inertial parameter $\beta$ is set at 0.01 and the Lipschitz constant is computed by backtracking. To make a fair comparison, we first run mGIST. The algorithm is terminated when the relative change of two consecutive objective function values is less than $10^{-5}$ or the number of iterations exceeds 1000. This termination condition is the same as in [19]. Then we run nmGIST, mAPG, nmAPG and IFB. These four algorithms are terminated when they achieve an equal or smaller objective function value than that by mGIST or the number of iterations exceeds 1000. We randomly choose 90% of the data as training data and the rest as test data. The experiment result is averaged over 10 runs. All algorithms are run on Matlab 2011a and Windows 7 with an Intel Core i3 2.53 GHz CPU and 4GB memory. The result is reported in Table 2. We also plot the curves of objective function values vs. iteration number and CPU time in Figure 1.

Table 2: Comparisons of APG, GIST and IFB on the sparse logistic regression problem. The quantities include number of iterations, averaged number of line searches in each iteration, computing time (in seconds) and test error. They are averaged over 10 runs.

| Method | #Iter. | #Line search | Time | test error |
|--------|--------|--------------|------|------------|
| mGIST  | 994    | 2.19         | 300.42 | 2.94% |
| nmGIST | 806    | 1.69         | 222.22 | 2.94% |
| IFB    | 635    | 2.59         | 215.82 | 2.96% |
| mAPG   | 175    | 2.99         | 133.23 | 2.93% |
| nmAPG  | 146    | 1.01         | 42.99  | 2.97% |

We have the following observations: (1) APG-type methods need much fewer iterations and less computing time than GIST and IFB to reach the same (or smaller) objective function values. As GIST is indeed a Proximal Gradient method (PG) and IFB is an extension of PG, this verifies that APG can indeed accelerate the convergence in practice. (2) nmAPG is faster than mAPG. We give two reasons: nmAPG avoids the computation of $\mathbf{v}_k$ in most of the time and reduces the number of line searches in each iteration. We mention that in mAPG line search is performed in both (11) and (12), while in nmAPG only the computation of $\mathbf{z}_{k+1}$ needs line search in every iteration. $\mathbf{v}_{k+1}$ is computed only when necessary. We note that the average number of line searches in nmAPG is nearly one. This means that (22) holds in most of the time. So we can trust that $\mathbf{z}_k$ can work well in most of the time and only in a few times $\mathbf{v}_k$ is computed to correct $\mathbf{z}_k$ and $\mathbf{y}_k$. On the other hand, nonmonotonicity allows for larger stepsizes, which results in fewer line searches.

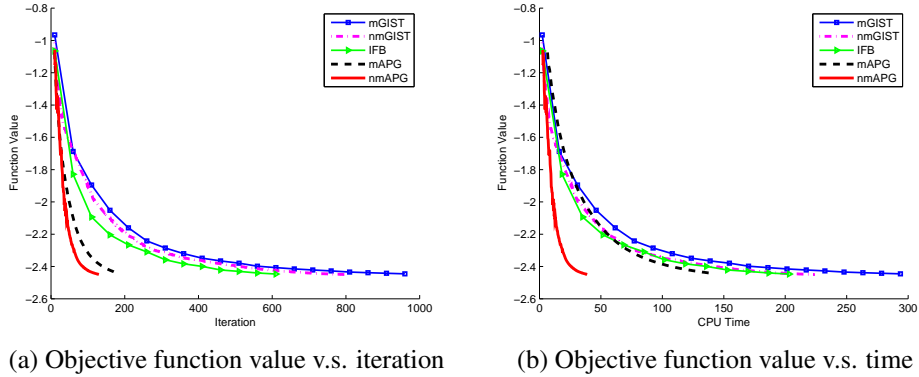

(a) Objective function value v.s. iteration        (b) Objective function value v.s. time

Figure 1: Compare the objective function value produced by APG, GIST and IFB.

## 5   Conclusions

In this paper, we propose two APG-type algorithms for efficiently solving general nonconvex nonsmooth problems, which are abundant in machine learning. We provide a detailed convergence analysis, showing that every accumulation point is a critical point for general nonconvex nonsmooth programs and the convergence rate is maintained at $O\left(\frac{1}{k^2}\right)$ for convex programs. Nonmonotone APG allows for larger stepsizes and needs less computation cost in each iteration and thus is faster than monotone APG in practice. Numerical experiments testify to the advantage of the two algorithms.

### Acknowledgments

Zhouchen Lin is supported by National Basic Research Program of China (973 Program) (grant no. 2015CB352502), National Natural Science Foundation (NSF) of China (grant nos. 61272341 and 61231002), and Microsoft Research Asia Collaborative Research Program. He is the corresponding author.

## Footnotes

[1]Except for the work under the KL assumption, convergence for nonconvex problems in this paper and the references of this paper means that every accumulation point is a critical point.

[2]For the detail of line search with Barzilai-Borwein initializtion please see Supplementary Materials.

[3]The proofs in this paper can be found in Supplementary Materials.

[4]For the details of difference please see Supplementary Materials.

[5] See Lemma 2 in Supplementary Materials.

[6] Please see Supplementary Materials for nonmonotone APG with line search.

[7]For the sake of space limitation we leave another experiment, Sparse PCA, in Supplementary Materials.

[8]http://www.public.asu.edu/ yje02/Software/GIST

[9]http://www.csie.ntu.edu.tw/cjlin/libsvmtools/datasets

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
