[Supplementary Material · APG_supp.pdf]

# Accelerated Proximal Gradient Methods for Nonconvex Programming

We consider a general problem:

$$\min_{\mathbf{x}\in\mathbb{R}^n} F(\mathbf{x}) = f(\mathbf{x}) + g(\mathbf{x}), \tag{1}$$

We mainly consider nonconvex $f$ and nonconvex nonsmooth $g$.

## 1 Preliminaries

### 1.1 Basic Assumptions

**Definition 1** *A function $g : \mathbb{R}^n \to (-\infty, +\infty]$ is said to be proper if dom $g \neq \emptyset$, where dom $g = \{\mathbf{x} \in \mathbb{R} : g(\mathbf{x}) < +\infty\}$. $g$ is lower semicontinuous at point $\mathbf{x}_0$ if*

$$\liminf_{x \to \mathbf{x}_0} g(\mathbf{x}) \geq g(\mathbf{x}_0). \tag{2}$$

In problem (1), we assume that $f$ is a proper function with Lipschitz continuous gradients and $g$ is proper and lower semicontinuous. We assume that $F(\mathbf{x})$ is coercive, i.e., $F$ is bounded from below and

$$F(\mathbf{x}) \to \infty \quad \text{when} \quad \|\mathbf{x}\| \to \infty, \tag{3}$$

where $\|\cdot\|$ is the $l_2$-norm.

### 1.2 Subdifferentials of Nonconvex and Nonsmooth Functions

**Definition 2** *[1, 2] Let $g$ be a proper and lower semicontinuous function.*

1. *For a given $\mathbf{x} \in$ dom $g$, the Frechet subdifferential of $g$ at $\mathbf{x}$, written as $\hat{\partial}g(\mathbf{x})$, is the set of all vectors $\mathbf{u} \in \mathbb{R}^n$ which satisfy*

$$\liminf_{\mathbf{y}\neq\mathbf{x},\mathbf{y}\to\mathbf{x}} \frac{g(\mathbf{y}) - g(\mathbf{x}) - \langle \mathbf{u}, \mathbf{y} - \mathbf{x}\rangle}{\|\mathbf{y} - \mathbf{x}\|} \geq 0. \tag{4}$$

2. *The limiting-subdifferential, or simply the subdifferential, of $g$ at $\mathbf{x} \in \mathbb{R}^n$, written as $\partial g(\mathbf{x})$, is defined through the following closure process*

$$\partial f(\mathbf{x}) := \{\mathbf{u} \in \mathbb{R}^n : \exists \mathbf{x}_k \to \mathbf{x}, g(\mathbf{x}_k) \to g(\mathbf{x}), \mathbf{u}_k \in \hat{\partial}g(\mathbf{x}_k) \to \mathbf{u}, k \to \infty\}. \tag{5}$$

**Proposition 1** *[1, 2]*

1. *In the nonsmooth context, the Fermat's rule remains unchanged: If $\mathbf{x} \in \mathbb{R}^n$ is a local minimizer of $g$, then $0 \in \partial g(\mathbf{x})$.*

2. *Let $(\mathbf{x}_k, \mathbf{u}_k)$ be a sequence such that $\mathbf{x}_k \to \mathbf{x}$, $\mathbf{u}_k \to \mathbf{u}$, $g(\mathbf{x}_k) \to g(\mathbf{x})$ and $\mathbf{u}_k \in \partial g(\mathbf{x}_k)$, then $\mathbf{u} \in \partial g(\mathbf{x})$.*

*3. If $f$ is a continuously differentiable function, then $\partial(f + g)(\mathbf{x}) = \nabla f(\mathbf{x}) + \partial g(\mathbf{x})$.*

Recall that points whose subdifferential contains 0 are called critical points.

## 1.3 Proximal Mapping

Let $g : \mathbb{R}^n \to (-\infty, +\infty]$ be a proper and lower semicontinuous function (it can be nonconvex). Given $\mathbf{x} \in \mathbb{R}^n$ and $\alpha > 0$, define the proximal mapping [1] as:

$$\text{prox}_{\alpha g}(\mathbf{x}) = \underset{\mathbf{u}}{\text{argmin}} \, g(\mathbf{u}) + \frac{1}{2\alpha}\|\mathbf{x} - \mathbf{u}\|^2. \tag{6}$$

When $g := \delta_X$, the indicator function of a nonempty and closed set $X$, defined as:

$$\delta_X(\mathbf{x}) = \begin{cases} 0, & \text{if } \mathbf{x} \in X, \\ \infty, & \text{otherwise,} \end{cases} \tag{7}$$

the proximal mapping reduces to the projection operator onto $X$, defined by

$$P_X(\mathbf{x}) = \{\mathbf{u} \in X : \mathbf{u} = \text{argmin} \, \|\mathbf{u} - \mathbf{x}\|\} \tag{8}$$

## 1.4 KL Inequality

**Definition 3** *[3, 2] A function $f : \mathbb{R}^n \to (-\infty, +\infty]$ is said to have the KL property at $\overline{\mathbf{u}} \in \text{dom}\partial f := \{\mathbf{x} \in \mathbb{R}^n : \partial f(\mathbf{u}) \neq \emptyset\}$ if there exists $\eta \in (0, +\infty]$, a neighborhood $U$ of $\overline{\mathbf{u}}$ and a function $\varphi \in \Phi_\eta$, such that for all*

$$\mathbf{u} \in U \bigcap \{\mathbf{u} \in \mathbb{R}^n : f(\overline{\mathbf{u}}) < f(\mathbf{u}) < f(\overline{\mathbf{u}}) + \eta\}, \tag{9}$$

*the following inequality holds*

$$\varphi'(f(\mathbf{u}) - f(\overline{\mathbf{u}}))dist(0, \partial f(\mathbf{u})) > 1, \tag{10}$$

*where $\Phi_\eta$ stands for a class of function $\varphi : [0, \eta) \to \mathbb{R}^+$ satisfying: (1) $\varphi$ is concave and $C^1$ on $(0, \eta)$; (2) $\varphi$ is continuous at 0, $\varphi(0) = 0$; and (3) $\varphi'(\mathbf{x}) > 0, \forall \mathbf{x} \in (0, \eta)$.*

**Lemma 1** *[2] Let $\Omega$ be a compact set and let $f : \mathbb{R}^n \to (-\infty, +\infty]$ be a proper and lower semicontinuous function. Assume that $f$ is constant on $\Omega$ and satisfies the KL property at each point of $\Omega$. Then there exists $\epsilon > 0$, $\eta > 0$ and $\varphi \in \Phi_\eta$, such that for all $\overline{\mathbf{u}}$ in $\Omega$ and all $\mathbf{u}$ in the following intersection*

$$\{\mathbf{u} \in \mathbb{R}^n : dist(\mathbf{u}, \Omega) < \epsilon\} \bigcap \{\mathbf{u} \in \mathbb{R}^n : f(\overline{\mathbf{u}}) < f(\mathbf{u}) < f(\overline{\mathbf{u}}) + \eta\}, \tag{11}$$

*the following inequality holds*

$$\varphi'(f(\mathbf{u}) - f(\overline{\mathbf{u}}))dist(0, \partial f(\mathbf{u})) > 1, \tag{12}$$

All semi-algebraic functions and subanalytic functions satisfy the KL property [3, 2]. So KL property is general enough. Typical examples include: real polynomial functions, logistic loss function $\log(1 + e^{-t})$, $\|\mathbf{x}\|_p$ $(p \geq 0)$, $\|\mathbf{x}\|_\infty$, indicator function of the positive semidefinite (PSD) cone, the Stiefel manifolds and the set of constant rank matrices.

## 2 Monotone APG

We summarize the monotone APG in Algorithm 1.

**Theorem 1** *Let $f$ be a proper function with Lipschitz continuous gradients and $g$ be proper and lower semicontinuous. For nonconvex $f$ and nonconvex nonsmooth $g$, assume that (3) holds. Then $\{\mathbf{x}_k\}$ and $\{\mathbf{v}_k\}$ generated by Algorithm 1 are bounded. Let $\mathbf{x}^*$ be any accumulation point of $\{\mathbf{x}_k\}$, we have $0 \in \partial F(\mathbf{x}^*)$.*

---

**Algorithm 1** monotone APG with fixed stepsize

---

Initialize $\mathbf{z}_1 = \mathbf{x}_1 = \mathbf{x}_0$, $t_1 = 1$, $t_0 = 0$, $\alpha_y < \frac{1}{L}$, $\alpha_x < \frac{1}{L}$.
**for** $k = 1, 2, 3, \cdots$ **do**

$$\mathbf{y}_k = \mathbf{x}_k + \frac{t_{k-1}}{t_k}(\mathbf{z}_k - \mathbf{x}_k) + \frac{t_{k-1} - 1}{t_k}(\mathbf{x}_k - \mathbf{x}_{k-1}), \tag{13}$$

$$\mathbf{z}_{k+1} = \text{prox}_{\alpha_y g}(\mathbf{y}_k - \alpha_y \nabla f(\mathbf{y}_k)), \tag{14}$$

$$\mathbf{v}_{k+1} = \text{prox}_{\alpha_x g}(\mathbf{x}_k - \alpha_x \nabla f(\mathbf{x}_k)), \tag{15}$$

$$t_{k+1} = \frac{\sqrt{4(t_k)^2 + 1} + 1}{2}, \tag{16}$$

$$\mathbf{x}_{k+1} = \begin{cases} \mathbf{z}_{k+1}, & \text{if } F(\mathbf{z}_{k+1}) \leq F(\mathbf{v}_{k+1}), \\ \mathbf{v}_{k+1}, & \text{otherwise.} \end{cases} \tag{17}$$

**end for**

---

**Proof** (15) in Algorithm 1 can be seen as

$$\mathbf{v}_{k+1} = \underset{\mathbf{x}}{\text{argmin}} \, \langle \nabla f(\mathbf{x}_k), \mathbf{x} - \mathbf{x}_k \rangle + \frac{1}{2\alpha_x}\|\mathbf{x} - \mathbf{x}_k\|^2 + g(\mathbf{x}). \tag{18}$$

So we have

$$\langle \nabla f(\mathbf{x}_k), \mathbf{v}_{k+1} - \mathbf{x}_k \rangle + \frac{1}{2\alpha_x}\|\mathbf{v}_{k+1} - \mathbf{x}_k\|^2 + g(\mathbf{v}_{k+1}) \leq g(\mathbf{x}_k). \tag{19}$$

From the Lipschitz continuous of $\nabla f$ we have

$$F(\mathbf{v}_{k+1}) \quad \leq \quad g(\mathbf{v}_{k+1}) + f(\mathbf{x}_k) + \langle \nabla f(\mathbf{x}_k), \mathbf{v}_{k+1} - \mathbf{x}_k \rangle + \frac{L}{2}\|\mathbf{v}_{k+1} - \mathbf{x}_k\|^2 \tag{20}$$

$$\leq \quad g(\mathbf{x}_k) - \langle \nabla f(\mathbf{x}_k), \mathbf{v}_{k+1} - \mathbf{x}_k \rangle - \frac{1}{2\alpha_x}\|\mathbf{v}_{k+1} - \mathbf{x}_k\|^2 \tag{21}$$

$$+ f(\mathbf{x}_k) + \langle \nabla f(\mathbf{x}_k), \mathbf{v}_{k+1} - \mathbf{x}_k \rangle + \frac{L}{2}\|\mathbf{v}_{k+1} - \mathbf{x}_k\|^2 \tag{22}$$

$$= \quad F(\mathbf{x}_k) - \left(\frac{1}{2\alpha_x} - \frac{L}{2}\right)\|\mathbf{v}_{k+1} - \mathbf{x}_k\|^2. \tag{23}$$

If $F(\mathbf{z}_{k+1}) \leq F(\mathbf{v}_{k+1})$, then

$$\mathbf{x}_{k+1} = \mathbf{z}_{k+1}, F(\mathbf{x}_{k+1}) = F(\mathbf{z}_{k+1}) \leq F(\mathbf{v}_{k+1}). \tag{24}$$

If $F(\mathbf{z}_{k+1}) > F(\mathbf{v}_{k+1})$, then

$$\mathbf{x}_{k+1} = \mathbf{v}_{k+1}, F(\mathbf{x}_{k+1}) = F(\mathbf{v}_{k+1}). \tag{25}$$

From (23), (24) and (25) we have

$$F(\mathbf{x}_{k+1}) \leq F(\mathbf{v}_{k+1}) \leq F(\mathbf{x}_k). \tag{26}$$

So

$$F(\mathbf{x}_{k+1}) \leq F(\mathbf{x}_1), F(\mathbf{v}_{k+1}) \leq F(\mathbf{x}_1) \tag{27}$$

for all $k$. From the assumption we know that $\{\mathbf{x}_k\}$ and $\{\mathbf{v}_k\}$ are bounded. Thus $\{\mathbf{x}_k\}$ has accumulation points. As $F(\mathbf{x}_k)$ is nonincreasing, $F$ has the same value at all the accumulation points. Let it be $F^*$. From (23) we have

$$\left(\frac{1}{2\alpha_x} - \frac{L}{2}\right)\|\mathbf{v}_{k+1} - \mathbf{x}_k\|^2 \leq F(\mathbf{x}_k) - F(\mathbf{v}_{k+1}) \leq F(\mathbf{x}_k) - F(\mathbf{x}_{k+1}). \tag{28}$$

Summing over $k = 1, 2, \cdots, \infty$, we have

$$\left(\frac{1}{2\alpha_x} - \frac{L}{2}\right)\sum_{k=1}^{\infty}\|\mathbf{v}_{k+1} - \mathbf{x}_k\|^2 \leq F(\mathbf{x}_1) - F^* < \infty, \tag{29}$$

From $\alpha_x < \frac{1}{L}$ we have

$$\|\mathbf{v}_{k+1} - \mathbf{x}_k\|^2 \to 0 \quad \text{as} \quad k \to \infty \tag{30}$$

From the optimality condition of (18) we have

$$0 \quad \in \quad \nabla f(\mathbf{x}_k) + \frac{1}{\alpha_x}(\mathbf{v}_{k+1} - \mathbf{x}_k) + \partial g(\mathbf{v}_{k+1}) \tag{31}$$

$$= \quad \nabla f(\mathbf{v}_{k+1}) + \nabla f(\mathbf{x}_k) - \nabla f(\mathbf{v}_{k+1}) + \frac{1}{\alpha_x}(\mathbf{v}_{k+1} - \mathbf{x}_k) + \partial g(\mathbf{v}_{k+1}). \tag{32}$$

So we have

$$-\nabla f(\mathbf{x}_k) + \nabla f(\mathbf{v}_{k+1}) - \frac{1}{\alpha_x}(\mathbf{v}_{k+1} - \mathbf{x}_k) \in \partial F(\mathbf{v}_{k+1}), \tag{33}$$

and

$$\left\| \nabla f(\mathbf{x}_k) - \nabla f(\mathbf{v}_{k+1}) + \frac{1}{\alpha_x}(\mathbf{v}_{k+1} - \mathbf{x}_k) \right\| \le \left( \frac{1}{\alpha_x} + L \right) \|\mathbf{v}_{k+1} - \mathbf{x}_k\| \to 0, \tag{34}$$

as $k \to \infty$.

Let $\mathbf{x}^*$ be any accumulation point of $\{\mathbf{x}_k\}$, say $\{\mathbf{x}_{k_j}\} \to \mathbf{x}^*$ as $j \to \infty$. From (30) we have $\{\mathbf{v}_{k_j+1}\} \to \mathbf{x}^*$ as $j \to \infty$. From (18) we have

$$\langle \nabla f(\mathbf{x}_{k_j}), \mathbf{v}_{k_j+1} - \mathbf{x}_{k_j} \rangle + \frac{1}{2\alpha_x}\|\mathbf{v}_{k_j+1} - \mathbf{x}_{k_j}\|^2 + g(\mathbf{v}_{k_j+1}) \tag{35}$$

$$\le \quad \langle \nabla f(\mathbf{x}_{k_j}), \mathbf{x}^* - \mathbf{x}_{k_j} \rangle + \frac{1}{2\alpha_x}\|\mathbf{x}^* - \mathbf{x}_{k_j}\|^2 + g(\mathbf{x}^*). \tag{36}$$

So

$$\limsup_{j \to \infty} g(\mathbf{v}_{k_j+1}) \le g(\mathbf{x}^*). \tag{37}$$

From the definition of lower semicontinuous of $g$ we have

$$\liminf_{j \to \infty} g(\mathbf{v}_{k_j+1}) \ge g(\mathbf{x}^*). \tag{38}$$

So we have

$$\lim_{j \to \infty} g(\mathbf{v}_{k_j+1}) = g(\mathbf{x}^*). \tag{39}$$

Because $f$ is continuously differentiable, we have

$$\lim_{j \to \infty} F(\mathbf{v}_{k_j+1}) = F(\mathbf{x}^*). \tag{40}$$

From $\{\mathbf{v}_{k_j+1}\} \to \mathbf{x}^*$, (40), (33), (34) and Proposition 1.2 we have

$$0 \in \partial F(\mathbf{x}^*). \tag{41}$$

∎

**Theorem 2** *Assume that $f$ and $g$ are convex and $\nabla f$ is Lipschitz continuous. Then $\{\mathbf{x}_k\}$ generated by algorithm 1 satisfies*

$$F(\mathbf{x}_{N+1}) - F(\mathbf{x}^*) \le \frac{2}{\alpha_y(N+1)^2}\|\mathbf{x}_0 - \mathbf{x}^*\|^2. \tag{42}$$

*where $\mathbf{x}^*$ is a global minimizer of $F(\mathbf{x})$.*

This theorem is quite similar to Theorem 5.1 in [4]. The proof is almost the same with [4]. We list the proof here only for the convenience of reader's reference.

**Proof** (14) in Algorithm 1 can be seen as

$$\mathbf{z}_{k+1} = \arg\min_{\mathbf{x}} \langle \nabla f(\mathbf{y}_k), \mathbf{x} - \mathbf{y}_k \rangle + \frac{1}{2\alpha_y}\|\mathbf{x} - \mathbf{y}_k\|^2 + g(\mathbf{x}). \tag{43}$$

From the optimality condition, we have

$$0 \in \nabla f(\mathbf{y}_k) + \frac{1}{\alpha_y}(\mathbf{z}_{k+1} - \mathbf{y}_k) + \partial g(\mathbf{z}_{k+1}). \tag{44}$$

From the convexity of $g$ we have

$$g(\mathbf{x}) - g(\mathbf{z}_{k+1}) \geq \left\langle -\nabla f(\mathbf{y}_k) - \frac{1}{\alpha_y}(\mathbf{z}_{k+1} - \mathbf{y}_k), \mathbf{x} - \mathbf{z}_{k+1} \right\rangle, \forall \mathbf{x}. \tag{45}$$

From the Lipschitz continuous of $\nabla f$ and convexity of $f$ we have

$$
\begin{align}
F(\mathbf{z}_{k+1}) \quad &\leq \quad g(\mathbf{z}_{k+1}) + f(\mathbf{y}_k) + \langle \nabla f(\mathbf{y}_k), \mathbf{z}_{k+1} - \mathbf{y}_k \rangle + \frac{L}{2}\|\mathbf{z}_{k+1} - \mathbf{y}_k\|^2 \tag{46} \\
&= \quad g(\mathbf{z}_{k+1}) + f(\mathbf{y}_k) + \langle \nabla f(\mathbf{y}_k), \mathbf{x} - \mathbf{y}_k \rangle + \langle \nabla f(\mathbf{y}_k), \mathbf{z}_{k+1} - \mathbf{x} \rangle \tag{47} \\
&\quad + \frac{L}{2}\|\mathbf{z}_{k+1} - \mathbf{y}_k\|^2 \tag{48} \\
&\leq \quad g(\mathbf{z}_{k+1}) + f(\mathbf{x}) + \langle \nabla f(\mathbf{y}_k), \mathbf{z}_{k+1} - \mathbf{x} \rangle + \frac{L}{2}\|\mathbf{z}_{k+1} - \mathbf{y}_k\|^2 \tag{49} \\
&\leq \quad g(\mathbf{x}) + \left\langle \nabla f(\mathbf{y}_k) + \frac{1}{\alpha_y}(\mathbf{z}_{k+1} - \mathbf{y}_k), \mathbf{x} - \mathbf{z}_{k+1} \right\rangle \tag{50} \\
&\quad + f(\mathbf{x}) + \langle \nabla f(\mathbf{y}_k), \mathbf{z}_{k+1} - \mathbf{x} \rangle + \frac{L}{2}\|\mathbf{z}_{k+1} - \mathbf{y}_k\|^2 \tag{51} \\
&= \quad F(\mathbf{x}) + \frac{1}{\alpha_y}\langle \mathbf{z}_{k+1} - \mathbf{y}_k, \mathbf{x} - \mathbf{z}_{k+1} \rangle + \frac{L}{2}\|\mathbf{z}_{k+1} - \mathbf{y}_k\|^2 \tag{52} \\
&= \quad F(\mathbf{x}) + \frac{1}{\alpha_y}\langle \mathbf{z}_{k+1} - \mathbf{y}_k, \mathbf{x} - \mathbf{y}_k + \mathbf{y}_k - \mathbf{z}_{k+1} \rangle + \frac{L}{2}\|\mathbf{z}_{k+1} - \mathbf{y}_k\|^2 \tag{53} \\
&= \quad F(\mathbf{x}) + \frac{1}{\alpha_y}\langle \mathbf{z}_{k+1} - \mathbf{y}_k, \mathbf{x} - \mathbf{y}_k \rangle - (\frac{1}{\alpha_y} - \frac{L}{2})\|\mathbf{z}_{k+1} - \mathbf{y}_k\|^2 \tag{54} \\
&\leq \quad F(\mathbf{x}) + \frac{1}{\alpha_y}\langle \mathbf{z}_{k+1} - \mathbf{y}_k, \mathbf{x} - \mathbf{y}_k \rangle - \frac{1}{2\alpha_y}\|\mathbf{z}_{k+1} - \mathbf{y}_k\|^2. \tag{55}
\end{align}
$$

Let $\mathbf{x} = \mathbf{x}_k$ and $\mathbf{x}^*$, we have

$$F(\mathbf{z}_{k+1}) - F(\mathbf{x}_k) \leq \frac{1}{\alpha_y}\langle \mathbf{z}_{k+1} - \mathbf{y}_k, \mathbf{x}_k - \mathbf{y}_k \rangle - \frac{1}{2\alpha_y}\|\mathbf{z}_{k+1} - \mathbf{y}_k\|^2, \tag{56}$$

$$F(\mathbf{z}_{k+1}) - F(\mathbf{x}^*) \leq \frac{1}{\alpha_y}\langle \mathbf{z}_{k+1} - \mathbf{y}_k, \mathbf{x}^* - \mathbf{y}_k \rangle - \frac{1}{2\alpha_y}\|\mathbf{z}_{k+1} - \mathbf{y}_k\|^2. \tag{57}$$

Multiplying (56) by $t_k - 1$ and adding (57) we have

$$t_k F(\mathbf{z}_{k+1}) - (t_k - 1)F(\mathbf{x}_k) - F(\mathbf{x}^*) \tag{58}$$

$$\leq \quad \frac{1}{\alpha_y}\langle \mathbf{z}_{k+1} - \mathbf{y}_k, (t_k - 1)(\mathbf{x}_k - \mathbf{y}_k) + \mathbf{x}^* - \mathbf{y}_k \rangle - \frac{t_k}{2\alpha_y}\|\mathbf{z}_{k+1} - \mathbf{y}_k\|^2. \tag{59}$$

So we have

$$t_k\left(F(\mathbf{z}_{k+1}) - F(\mathbf{x}^*)\right) - (t_k - 1)\left(F(\mathbf{x}_k) - F(\mathbf{x}^*)\right) \tag{60}$$

$$\leq \quad \frac{1}{\alpha_y}\langle \mathbf{z}_{k+1} - \mathbf{y}_k, (t_k - 1)(\mathbf{x}_k - \mathbf{y}_k) + \mathbf{x}^* - \mathbf{y}_k \rangle - \frac{t_k}{2\alpha_y}\|\mathbf{z}_{k+1} - \mathbf{y}_k\|^2. \tag{61}$$

Multiplying both sides by $t_k$ and using $(t_k)^2 - t_k = (t_{k-1})^2$ from (16) we have

$$(t_k)^2\left(F(\mathbf{z}_{k+1}) - F(\mathbf{x}^*)\right) - (t_{k-1})^2\left(F(\mathbf{x}_k) - F(\mathbf{x}^*)\right) \tag{62}$$

$$\leq \quad \frac{1}{\alpha_y}\langle t_k(\mathbf{z}_{k+1} - \mathbf{y}_k), (t_k - 1)(\mathbf{x}_k - \mathbf{y}_k) + \mathbf{x}^* - \mathbf{y}_k \rangle - \frac{1}{2\alpha_y}\|t_k(\mathbf{z}_{k+1} - \mathbf{y}_k)\|^2 \tag{63}$$

$$= \quad \frac{1}{\alpha_y}\langle t_k(\mathbf{z}_{k+1} - \mathbf{y}_k), (t_k - 1)\mathbf{x}_k - t_k\mathbf{y}_k + \mathbf{x}^* \rangle - \frac{1}{2\alpha_y}\|t_k(\mathbf{z}_{k+1} - \mathbf{y}_k)\|^2 \tag{64}$$

$$= \quad \frac{1}{2\alpha_y}\left(\|(t_k - 1)\mathbf{x}_k - t_k\mathbf{y}_k + \mathbf{x}^*\|^2 - \|(t_k - 1)\mathbf{x}_k - t_k\mathbf{z}_{k+1} + \mathbf{x}^*\|^2\right). \tag{65}$$

Define
$$U_{k+1} = t_k \mathbf{z}_{k+1} - (t_k - 1)\mathbf{x}_k - \mathbf{x}^*. \tag{66}$$

Let
$$U_k = t_{k-1}\mathbf{z}_k - (t_{k-1} - 1)\mathbf{x}_{k-1} - \mathbf{x}^* = t_k \mathbf{y}_k - (t_k - 1)\mathbf{x}_k - \mathbf{x}^*. \tag{67}$$

We have
$$\mathbf{y}_k = \frac{t_{k-1}\mathbf{z}_k - (t_{k-1} - 1)\mathbf{x}_{k-1} + (t_k - 1)\mathbf{x}_k}{t_k} \tag{68}$$

$$= \mathbf{x}_k + \frac{t_{k-1}}{t_k}(\mathbf{z}_k - \mathbf{x}_k) + \frac{t_{k-1} - 1}{t_k}(\mathbf{x}_k - \mathbf{x}_{k-1}), \tag{69}$$

which is the same with (13) in Algorithm 1. So we have
$$(t_k)^2 \left( F(\mathbf{z}_{k+1}) - F(\mathbf{x}^*) \right) - (t_{k-1})^2 \left( F(\mathbf{x}_k) - F(\mathbf{x}^*) \right) \tag{70}$$

$$\leq \frac{1}{2\alpha_y} \left( \|U_k\|^2 - \|U_{k+1}\|^2 \right). \tag{71}$$

If $F(\mathbf{z}_{k+1}) \leq F(\mathbf{v}_{k+1})$, then $\mathbf{x}_{k+1} = \mathbf{z}_{k+1}$. So
$$(t_k)^2 \left( F(\mathbf{x}_{k+1}) - F(\mathbf{x}^*) \right) - (t_{k-1})^2 \left( F(\mathbf{x}_k) - F(\mathbf{x}^*) \right) \tag{72}$$

$$= (t_k)^2 \left( F(\mathbf{z}_{k+1}) - F(\mathbf{x}^*) \right) - (t_{k-1})^2 \left( F(\mathbf{x}_k) - F(\mathbf{x}^*) \right) \tag{73}$$

$$\leq \frac{1}{2\alpha_y} \left( \|U_k\|^2 - \|U_{k+1}\|^2 \right). \tag{74}$$

If $F(\mathbf{z}_{k+1}) > F(\mathbf{v}_{k+1})$, then $\mathbf{x}_{k+1} = \mathbf{v}_{k+1}$. So
$$(t_k)^2 \left( F(\mathbf{x}_{k+1}) - F(\mathbf{x}^*) \right) - (t_{k-1})^2 \left( F(\mathbf{x}_k) - F(\mathbf{x}^*) \right) \tag{75}$$

$$\leq (t_k)^2 \left( F(\mathbf{z}_{k+1}) - F(\mathbf{x}^*) \right) - (t_{k-1})^2 \left( F(\mathbf{x}_k) - F(\mathbf{x}^*) \right) \tag{76}$$

$$\leq \frac{1}{2\alpha_y} \left( \|U_k\|^2 - \|U_{k+1}\|^2 \right). \tag{77}$$

Summing over $k = 1, \cdots, N$, we have
$$(t_N)^2 \left( F(\mathbf{x}_{N+1}) - F(\mathbf{x}^*) \right) \tag{78}$$

$$= (t_N)^2 \left( F(\mathbf{x}_{N+1}) - F(\mathbf{x}^*) \right) - (t^0)^2 \left( F(\mathbf{x}_1) - F(\mathbf{x}^*) \right) \tag{79}$$

$$\leq \frac{1}{2\alpha_y} \left( \|U_1\|^2 - \|U_{N+1}\|^2 \right) \tag{80}$$

$$\leq \frac{1}{2\alpha_y} \|U_1\|^2 \tag{81}$$

$$= \frac{1}{2\alpha_y} \|\mathbf{x}_0 - \mathbf{x}^*\|^2. \tag{82}$$

From (16) we can easily have that $t_k \geq \frac{k+1}{2}$. So we have
$$F(\mathbf{x}_{N+1}) - F(\mathbf{x}^*) \leq \frac{2}{\alpha_y(N+1)^2} \|\mathbf{x}_0 - \mathbf{x}^*\|^2. \tag{83}$$

∎

**Theorem 3** *Let $f$ be a proper function with Lipschitz continuous gradients and $g$ be proper and lower semicontinuous. For nonconvex $f$ and nonconvex nonsmooth $g$, assume that (3) holds. If we further assume that $f$ and $g$ satisfy the KL property, and the desingularising function has the form of $\varphi(t) = \frac{C}{\theta}t^\theta$ for some $C > 0$, $\theta \in (0, 1]$, then*

1. *If $\theta = 1$, then there exists $k_1$ such that $F(\mathbf{x}_k) = F^*$ for all $k > k_1$ and the algorithm terminates in finite steps.*

2. *If $\theta \in [\frac{1}{2}, 1)$, then there exists $k_2$ such that for all $k > k_2$,*

$$F(\mathbf{x}_k) - F^* \leq \left( \frac{d_1 C^2}{1 + d_1 C^2} \right)^{k-k_2} r_{k_2}. \tag{84}$$

*3. If $\theta \in (0, \frac{1}{2})$, then there exists $k_3$ such that for all $k > k_3$,*

$$F(\mathbf{x}_k) - F^* \leq \left( \frac{C}{(k - k_3)d_2(1 - 2\theta)} \right)^{\frac{1}{1-2\theta}}, \tag{85}$$

*where $F^*$ is the same function value at all the accumulation points of $\{\mathbf{x}_k\}$, $r_k = F(\mathbf{v}_k) - F^*$, $d_1 = \left( \frac{1}{\alpha_x} + L \right)^2 / \left( \frac{1}{2\alpha_x} - \frac{L}{2} \right)$, $d_2 = \min \left\{ \frac{1}{2d_1 C}, \frac{C}{1-2\theta} \left( 2^{\frac{2\theta-1}{2\theta-2}} - 1 \right) r_0^{2\theta-1} \right\}$*

This theorem is similar to Theorem 4 in [5] and the proof is almost the same with [5]. We will discuss the difference later.

**Proof** From (23) and (26) we have

$$F(\mathbf{v}_{k+1}) \leq F(\mathbf{x}_k) - \left( \frac{1}{2\alpha_x} - \frac{L}{2} \right) \|\mathbf{v}_{k+1} - \mathbf{x}_k\|^2 \tag{86}$$

$$\leq F(\mathbf{v}_k) - \left( \frac{1}{2\alpha_x} - \frac{L}{2} \right) \|\mathbf{v}_{k+1} - \mathbf{x}_k\|^2. \tag{87}$$

From (34) we have

$$\text{dist}(0, \partial F(\mathbf{v}_{k+1})) \leq \left( \frac{1}{\alpha_x} + L \right) \|\mathbf{v}_{k+1} - \mathbf{x}_k\|. \tag{88}$$

From (30) we know that $\{\mathbf{x}_k\}$ and $\{\mathbf{v}_k\}$ have the same accumulation points. Let $\Omega$ be the set that contains all the accumulation points of $\{\mathbf{x}_k\}$ (also $\{\mathbf{v}_k\}$). Because $F(\mathbf{v}_k)$ is nonincreasing, $F$ has the same value at all the accumulation points in $\Omega$. Let it be $F^*$. So we have

$$F(\mathbf{v}_k) \geq F^*, F(\mathbf{v}_k) \to F^*. \tag{89}$$

If there exists $\overline{k}$ such that $F(\mathbf{v}^{\overline{k}}) = F^*$, then $F(\mathbf{v}^{\overline{k}}) = F(\mathbf{v}^{\overline{k}+1}) = \cdots = F^*$. So $\|\mathbf{v}^{\overline{k}+1} - \mathbf{x}^{\overline{k}}\| = \|\mathbf{v}^{\overline{k}+2} - \mathbf{x}^{\overline{k}+1}\| = \cdots = 0$. The conclusion holds. If $F(\mathbf{v}_k) > F^*$ for all $k$, then from $F(\mathbf{v}_k) \to F^*$ we know that there exists $\hat{k}_1$ such that $F(\mathbf{v}_k) < F^* + \eta$ whenever $k > \hat{k}_1$. On the other hand, because $\text{dist}(\mathbf{v}_k, \Omega) \to 0$, there exists $\hat{k}_2$ such that $\text{dist}(\mathbf{v}_k, \Omega) < \varepsilon$ whenever $k > \hat{k}_2$. Let $k > k_0 = \max\{\hat{k}_1, \hat{k}_2\}$, we have

$$\mathbf{v}_k \in \{\mathbf{v}, \text{dist}(\mathbf{v}, \Omega) \leq \varepsilon\} \bigcap [F^* < F(\mathbf{v}) < F^* + \eta]. \tag{90}$$

From the uniform KL property in Lemma 1, there exists a concave function $\varphi$ such that

$$\varphi'(F(\mathbf{v}_k) - F^*)\text{dist}(0, \partial F(\mathbf{v}_k)) \geq 1. \tag{91}$$

Define $r_k = F(\mathbf{v}_k) - F^*$. We suppose that $r_k > 0$ for all $k$. Otherwise $F(\mathbf{v}_k) = F(\mathbf{v}_{k+1}) = \cdots = F^*$ and the algorithm terminates in finite steps. By supposing this (91) holds.

From (88), (91) and (87) we have

$$1 \leq [\varphi'(F(\mathbf{v}_k) - F^*)\text{dist}(0, \partial F(\mathbf{v}_k))]^2 \tag{92}$$

$$\leq [\varphi'(r_k)]^2 \left( \frac{1}{\alpha_x} + L \right)^2 \|\mathbf{v}_k - \mathbf{x}_{k-1}\|^2 \tag{93}$$

$$\leq [\varphi'(r_k)]^2 \left( \frac{1}{\alpha_x} + L \right)^2 \frac{F(\mathbf{v}_{k-1}) - F(\mathbf{v}_k)}{\left( \frac{1}{2\alpha_x} - \frac{L}{2} \right)} \tag{94}$$

$$= d_1[\varphi'(r_k)]^2 (r_{k-1} - r_k), \tag{95}$$

for all $k > k_0$, where $d_1 = \left( \frac{1}{\alpha_x} + L \right)^2 / \left( \frac{1}{2\alpha_x} - \frac{L}{2} \right)$. Because $\varphi$ has the form of $\varphi(t) = \frac{C}{\theta} t^\theta$, we have $\varphi'(t) = Ct^{\theta-1}$. So (95) becomes

$$1 \leq d_1 C^2 r_k^{2\theta-2} (r_{k-1} - r_k). \tag{96}$$

1. Case $\theta = 1$.

In this case, (96) becomes

$$1 \leq d_1 C^2 (r_k - r_{k+1}). \tag{97}$$

Because $r_k \to 0$ and $d_1 > 0$, $C > 0$, this is a contradiction. So there exists $k_1$ such that $r_k = 0$ for all $k > k_1$. The algorithm terminates in finite steps.

2. Case $\theta \in [\frac{1}{2}, 1)$.

In this case, $0 < 2 - 2\theta \leq 1$. As $r_k \to 0$, there exists $\hat{k}_3$ such that $r_k^{2-2\theta} \geq r_k$ for all $k > \hat{k}_3$. (96) becomes

$$r_k \leq d_1 C^2 (r_{k-1} - r_k). \tag{98}$$

So we have

$$r_k \leq \frac{d_1 C^2}{1 + d_1 C^2} r_{k-1}, \tag{99}$$

for all $k_2 > max\{k_0, \hat{k}_3\}$ and

$$r_k \leq \left( \frac{d_1 C^2}{1 + d_1 C^2} \right)^{k-k_2} r_{k_2}. \tag{100}$$

So we have

$$F(\mathbf{x}_k) - F^* \leq F(\mathbf{v}_k) - F^* = r_k \leq \left( \frac{d_1 C^2}{1 + d_1 C^2} \right)^{k-k_2} r_{k_2}. \tag{101}$$

3. Case $\theta \in (0, \frac{1}{2})$.

In this case, $2\theta - 2 \in (-2, -1)$, $2\theta - 1 \in (-1, 0)$. As $r_{k-1} > r_k$, we have $r_{k-1}^{2\theta-2} < r_k^{2\theta-2}$ and $r_0^{2\theta-1} < \cdots < r_{k-1}^{2\theta-1} < r_k^{2\theta-1}$

Define $\phi(t) = \frac{C}{1-2\theta} t^{2\theta-1}$, then $\phi'(t) = -Ct^{2\theta-2}$.

If $r_k^{2\theta-2} \leq 2r_{k-1}^{2\theta-2}$, then

$$\phi(r_k) - \phi(r_{k-1}) = \int_{r_{k-1}}^{r_k} \phi'(t)dt = C \int_{r_k}^{r_{k-1}} t^{2\theta-2}dt \tag{102}$$

$$\geq C(r_{k-1} - r_k)r_{k-1}^{2\theta-2} \geq \frac{C}{2}(r_{k-1} - r_k)r_k^{2\theta-2} \tag{103}$$

$$\geq \frac{1}{2d_1 C}. \tag{104}$$

for all $k > k_0$.

If $r_k^{2\theta-2} \geq 2r_{k-1}^{2\theta-2}$, then $r_k^{2\theta-1} \geq 2^{\frac{2\theta-1}{2\theta-2}} r_{k-1}^{2\theta-1}$.

$$\phi(r_k) - \phi(r_{k-1}) = \frac{C}{1-2\theta}(r_k^{2\theta-1} - r_{k-1}^{2\theta-1}) \tag{105}$$

$$\geq \frac{C}{1-2\theta}(2^{\frac{2\theta-1}{2\theta-2}} - 1)r_{k-1}^{2\theta-1} \tag{106}$$

$$= qr_{k-1}^{2\theta-1} \geq qr_0^{2\theta-1}. \tag{107}$$

where $q = \frac{C}{1-2\theta}(2^{\frac{2\theta-1}{2\theta-2}} - 1)$. Let $d_2 = \min\{\frac{1}{2d_1 C}, qr_0^{2\theta-1}\}$, we have

$$\phi(r_k) - \phi(r_{k-1}) \geq d_2, \tag{108}$$

for all $k > k_0$ and

$$\phi(r_k) \geq \phi(r_k) - \phi(r_{k_0}) \geq \sum_{i=k_0+1}^{k} \phi(r_i) - \phi(r_{i-1}) \geq (k - k_0)d_2. \tag{109}$$

So we have

$$r_k^{2\theta-1} \geq \frac{(k-k_0)d_2(1-2\theta)}{C}, \tag{110}$$

and

$$r_k \leq \left(\frac{C}{(k-k_0)d_2(1-2\theta)}\right)^{\frac{1}{1-2\theta}}. \tag{111}$$

Let $k_3 = k_0$ we have

$$F(\mathbf{x}_k) - F^* \leq F(\mathbf{v}_k) - F^* = r_k \leq \left(\frac{C}{(k-k_3)d_2(1-2\theta)}\right)^{\frac{1}{1-2\theta}}, \tag{112}$$

which completes the proof. ∎

Difference with the conditions in [5]:

[5] considered general descent method with the conditions:

$$F(\mathbf{x}_{k+1}) \leq F(\mathbf{x}_k) - \alpha\|\mathbf{x}_{k+1} - \mathbf{x}_k\|, \tag{113}$$

and

$$\|\partial F(\mathbf{x}_{k+1})\| \leq \beta\|\mathbf{x}_{k+1} - \mathbf{x}_k\| \tag{114}$$

Proximal gradient method is a typical example satisfying these conditions. However, to make the proximal gradient method both accelerate and converge, we introduce the intermediate variables $\mathbf{y}_k$, $\mathbf{v}_k$ and $\mathbf{z}_k$. This makes our algorithm more complex and the conditions satisfied by our algorithm becomes

$$F(\mathbf{x}_{k+1}) \leq F(\mathbf{x}_k) - \alpha\|\mathbf{v}_{k+1} - \mathbf{x}_k\|, F(\mathbf{v}_{k+1}) \leq F(\mathbf{v}_k) - \alpha\|\mathbf{v}_{k+1} - \mathbf{x}_k\| \tag{115}$$

and

$$\|\partial F(\mathbf{v}_{k+1})\| \leq \beta\|\mathbf{v}_{k+1} - \mathbf{x}_k\| \tag{116}$$

The intermediate variable $\mathbf{v}$ makes the main difference. As a result, under the conditions of (113) and (114), a useful conclusion of finite length of $\{\mathbf{x}\}$: $\sum_{i=k}^{\infty}\|\mathbf{x}_{k+1} - \mathbf{x}_k\| < \infty$ can be achieved and $\{\mathbf{x}_k\}$ is a converged sequence. Accordingly, the convergence rate for $\|\mathbf{x}_k - \mathbf{x}^*\|$ can be obtained. By contrast, our algorithm can only get $\sum_{i=1}^{\infty}\|\mathbf{v}_{k+1} - \mathbf{x}_k\| < \infty$. Neither the convergence rate for $\|\mathbf{x}_k - \mathbf{x}^*\|$ nor $\{\mathbf{x}_k\}$ is a converged sequence can be obtained.

## 2.1 Backtracking Line Search with Barzilai-Borwein Initializtion

In order to allow for larger step sizes, and thus faster convergence, we can use line search initialized with the Barzilai-Borwein (BB) rule to compute the step sizes $\alpha_x$ and $\alpha_y$. For $\alpha_x$, we choose the smallest $h \geq 0$, such that $\mathbf{v}_{k+1}$ computed by

$$\alpha_x = \alpha_{x,0}\rho^h, \tag{117}$$

$$\mathbf{v}_{k+1} = \text{prox}_{\alpha_x g}(\mathbf{x}_k - \alpha_x \nabla f(\mathbf{x}_k)), \tag{118}$$

satisfies

$$F(\mathbf{v}_{k+1}) \leq F(\mathbf{x}_k) - \delta\|\mathbf{v}_{k+1} - \mathbf{x}_k\|^2, \tag{119}$$

where

$$\alpha_{x,0} = \left|\frac{(\mathbf{s}_k)^T \mathbf{s}_k}{(\mathbf{s}_k)^T \mathbf{r}_k}\right| \quad \text{or} \quad \alpha_{x,0} = \left|\frac{(\mathbf{s}_k)^T \mathbf{r}_k}{(\mathbf{r}_k)^T \mathbf{r}_k}\right|, \tag{120}$$

$$\mathbf{s}_k = \mathbf{v}_k - \mathbf{x}_{k-1}, \mathbf{r}_k = \nabla f(\mathbf{v}_k) - \nabla f(\mathbf{x}_{k-1}), \tag{121}$$

$0 < \rho < 1$ and $\delta > 0$ is a small constant. Such $\mathbf{v}_{k+1}$ satisfying (119) can be found in finite iterations. In the worst case, $\alpha_x$ will be reduced until $\alpha_x < \frac{1}{L}$. $\alpha_y$ can also use this strategy. It is remarkable that Theorem 1 still holds when line search is used for $\alpha_y$ and $\alpha_x$. To make Theorem 2 hold, instead of (119), we should use the following condition for $\alpha_y$:

$$f(\mathbf{z}_{k+1}) \leq f(\mathbf{y}_k) + \langle \nabla f(\mathbf{y}_k), \mathbf{z}_{k+1} - \mathbf{y}_k \rangle + \frac{1}{2\alpha_y}\|\mathbf{z}_{k+1} - \mathbf{y}_k\|^2. \tag{122}$$

We summarize the monotone APG with line search in Algorithm 2.

---

**Algorithm 2** monotone APG with line search

---

Initialize $\mathbf{z}_1 = \mathbf{x}_1 = \mathbf{x}_0, t_1 = 1, t_0 = 0, \delta > 0, \rho < 1$.
**for** $k = 1, 2, 3, \cdots$ **do**

$$\mathbf{y}_k = \mathbf{x}_k + \frac{t_{k-1}}{t_k}(\mathbf{z}_k - \mathbf{x}_k) + \frac{t_{k-1} - 1}{t_k}(\mathbf{x}_k - \mathbf{x}_{k-1}), \tag{123}$$

$$\mathbf{s}_k = \mathbf{z}_k - \mathbf{y}_{k-1}, \mathbf{r}_k = \nabla f(\mathbf{z}_k) - \nabla f(\mathbf{y}_{k-1}), \tag{124}$$

$$\alpha_y = \frac{(\mathbf{s}_k)^T \mathbf{s}_k}{(\mathbf{s}_k)^T \mathbf{r}_k} \quad or \quad \alpha_y = \frac{(\mathbf{s}_k)^T \mathbf{r}_k}{(\mathbf{r}_k)^T \mathbf{r}_k}, \tag{125}$$

$$\mathbf{s}_k = \mathbf{v}_k - \mathbf{x}_{k-1}, \mathbf{r}_k = \nabla f(\mathbf{v}_k) - \nabla f(\mathbf{x}_{k-1}), \tag{126}$$

$$\alpha_x = \frac{(\mathbf{s}_k)^T \mathbf{s}_k}{(\mathbf{s}_k)^T \mathbf{r}_k} \quad or \quad \alpha_x = \frac{(\mathbf{s}_k)^T \mathbf{r}_k}{(\mathbf{r}_k)^T \mathbf{r}_k}. \tag{127}$$

Repeat

$$\mathbf{z}_{k+1} = \text{prox}_{\alpha_y g}(\mathbf{y}_k - \alpha_y \nabla f(\mathbf{y}_k)), \tag{128}$$

$$\alpha_y = \alpha_y \times \rho, \tag{129}$$

until $\quad F(\mathbf{z}_{k+1}) \leq F(\mathbf{y}_k) - \delta \|\mathbf{z}_{k+1} - \mathbf{y}_k\|^2$.
Repeat

$$\mathbf{v}_{k+1} = \text{prox}_{\alpha_x g}(\mathbf{x}_k - \alpha_x \nabla f(\mathbf{x}_k)), \tag{130}$$

$$\alpha_x = \alpha_x \times \rho, \tag{131}$$

until $\quad F(\mathbf{v}_{k+1}) \leq F(\mathbf{x}_k) - \delta \|\mathbf{v}_{k+1} - \mathbf{x}_k\|^2$.

$$t_{k+1} = \frac{\sqrt{4(t_k)^2 + 1} + 1}{2}, \tag{132}$$

$$\mathbf{x}_{k+1} = \begin{cases} \mathbf{z}_{k+1}, & \text{if } F(\mathbf{z}_{k+1}) \leq F(\mathbf{v}_{k+1}), \\ \mathbf{v}_{k+1}, & \text{otherwise.} \end{cases} \tag{133}$$

**end for**

---

**Corollary 1** *Let $f$ be a proper function with Lipschitz continuous gradients and $g$ be proper and lower semicontinuous. For nonconvex $f$ and nonconvex nonsmooth $g$, assume that (3) holds, then $\{\mathbf{x}_k\}$ and $\{\mathbf{v}_k\}$ generated by Algorithm 2 are bounded. Let $\mathbf{x}^*$ be any accumulation point of $\{\mathbf{x}_k\}$, we have $0 \in \partial F(\mathbf{x}^*)$.*

**Proof** From (23) and similar deduction we know that such $\alpha_y$ and $\alpha_x$ satisfying

$$\mathbf{v}_{k+1} = \text{prox}_{\alpha_x g}(\mathbf{x}_k - \alpha_x \nabla f(\mathbf{x}_k)), \tag{134}$$

$$F(\mathbf{v}_{k+1}) \leq F(\mathbf{x}_k) - \delta \|\mathbf{v}_{k+1} - \mathbf{x}_k\|^2, \tag{135}$$

$$\mathbf{z}_{k+1} = \text{prox}_{\alpha_y g}(\mathbf{y}_k - \alpha_y \nabla f(\mathbf{y}_k)), \tag{136}$$

$$F(\mathbf{z}_{k+1}) \leq F(\mathbf{y}_k) - \delta \|\mathbf{z}_{k+1} - \mathbf{y}_k\|^2, \tag{137}$$

exist, e.g., when they are reduced until $\alpha_x < \frac{1}{L}$ and $\alpha_y < \frac{1}{L}$. So the line search can be terminated in finite iterations. Similar to Theorem 1 we can have the conclusion. ∎

## 3 Nonmonotone APG

We summarize the nonmonotone APG in Algorithm 3 and nonmonotone APG with line search in Algorithm 4.

Different from (121), we use the following $\mathbf{s}_k$ and $\mathbf{r}_k$ to initialize $\alpha_{x,0}$ when line search is used:

$$\mathbf{s}_k = \mathbf{x}_k - \mathbf{y}_{k-1}, \mathbf{r}_k = \nabla f(\mathbf{x}_k) - \nabla f(\mathbf{y}_{k-1}), \tag{138}$$

---

**Algorithm 3** nonmonotone APG with fixed stepsize

---

Initialize $\mathbf{z}_1 = \mathbf{x}_1 = \mathbf{x}_0$, $t_1 = 1$, $t_0 = 0$, $\eta \in [0,1)$, $\delta > 0$, $c_1 = F(\mathbf{x}_1)$, $q_1 = 1$, $\alpha_x < \frac{1}{L}$, $\alpha_y < \frac{1}{L}$.
**for** $k = 1, 2, 3, \cdots$ **do**

$$\mathbf{y}_k = \mathbf{x}_k + \frac{t_{k-1}}{t_k}(\mathbf{z}_k - \mathbf{x}_k) + \frac{t_{k-1} - 1}{t_k}(\mathbf{x}_k - \mathbf{x}_{k-1}), \tag{140}$$

$$\mathbf{z}_{k+1} = \text{prox}_{\alpha_y g}(\mathbf{y}_k - \alpha_y \nabla f(\mathbf{y}_k)), \tag{141}$$

**if** $F(\mathbf{z}_{k+1}) \leq c_k - \delta\|\mathbf{z}_{k+1} - \mathbf{y}_k\|^2$ **then**

$$\mathbf{x}_{k+1} = \mathbf{z}_{k+1}. \tag{142}$$

**else**

$$\mathbf{v}_{k+1} = \text{prox}_{\alpha_x g}(\mathbf{x}_k - \alpha_x \nabla f(\mathbf{x}_k)), \tag{143}$$

$$\mathbf{x}_{k+1} = \begin{cases} \mathbf{z}_{k+1}, & \text{if } F(\mathbf{z}_{k+1}) \leq F(\mathbf{v}_{k+1}), \\ \mathbf{v}_{k+1}, & \text{otherwise.} \end{cases} \tag{144}$$

**end if**

$$t_{k+1} = \frac{\sqrt{4(t_k)^2 + 1} + 1}{2}, \tag{145}$$

$$q_{k+1} = \eta q_k + 1, \tag{146}$$

$$c_{k+1} = \frac{\eta q_k c_k + F(\mathbf{x}_{k+1})}{q_{k+1}}. \tag{147}$$

**end for**

---

This is because in nonmonotone APG, $\mathbf{v}_k$ is not computed in every iteration. So $\alpha_{x,0}$ should be initialized only by the recent and existing information and we should also avoid additional calculations of $\nabla f$. Similarly, $\alpha_{y,0}$ is initialized by the following $\mathbf{s}_k$ and $\mathbf{r}_k$:

$$\mathbf{s}_k = \mathbf{y}_k - \mathbf{y}_{k-1}, \mathbf{r}_k = \nabla f(\mathbf{y}_k) - \nabla f(\mathbf{y}_{k-1}). \tag{139}$$

In practice, the line search for $\alpha_y$ is terminated when $F(\mathbf{z}_{k+1}) \leq F(\mathbf{y}_k) - \delta\|\mathbf{z}_{k+1} - \mathbf{y}_k\|^2$ or $F(\mathbf{z}_{k+1}) \leq c_k - \delta\|\mathbf{z}_{k+1} - \mathbf{y}_k\|^2$ holds. Theoretically, we should use (122) as the stopping criteria to make Theorem 2 hold. However, it needs more line searches.

**Lemma 2** *In Algorithms 3 and 4, we have*

$$F(\mathbf{x}_k) \leq c_k \leq A_k, A_k = \frac{\sum_{i=1}^{k} F(\mathbf{x}_i)}{k}, \tag{162}$$

*and there exists $\alpha_x$ such that*

$$\mathbf{v}_{k+1} = prox_{\alpha_x g}(\mathbf{x}_k - \alpha_x \nabla f(\mathbf{x}_k)) \tag{163}$$

*satisfies*

$$F(\mathbf{v}_{k+1}) \leq c_k - \delta\|\mathbf{v}_{k+1} - \mathbf{x}_k\|^2, \tag{164}$$

*where $\delta$ is any small positive constant.*

The proof of (162) is similar to lemma 1.1 in [6]. Here we list the proof only for the sake of completeness.

**Proof** We prove by induction. For $k = 1$, $c_1 = F(\mathbf{x}_1)$. From (18)-(23) we know that $\alpha_x < \frac{1}{L}$ satisfies

$$F(\mathbf{v}_2) \leq c_1 - \delta\|\mathbf{v}_2 - \mathbf{x}_1\|, \tag{165}$$

---

**Algorithm 4** nonmonotone APG with line search

---

Initialize $\mathbf{z}_1 = \mathbf{x}_1 = \mathbf{x}_0$, $t_1 = 1$, $t_0 = 0$, $\eta \in [0, 1)$, $\delta > 0$, $\rho < 1$, $c_1 = F(\mathbf{x}_1)$, $q_1 = 1$.
**for** $k = 1, 2, 3, \cdots$ **do**

$$\mathbf{y}_k = \mathbf{x}_k + \frac{t_{k-1}}{t_k}(\mathbf{z}_k - \mathbf{x}_k) + \frac{t_{k-1} - 1}{t_k}(\mathbf{x}_k - \mathbf{x}_{k-1}), \tag{148}$$

$$\mathbf{s}_k = \mathbf{y}_k - \mathbf{y}_{k-1}, \mathbf{r}_k = \nabla f(\mathbf{y}_k) - \nabla f(\mathbf{y}_{k-1}), \tag{149}$$

$$\alpha_y = \frac{(\mathbf{s}_k)^T \mathbf{s}_k}{(\mathbf{s}_k)^T \mathbf{r}_k} \quad or \quad \alpha_y = \frac{(\mathbf{s}_k)^T \mathbf{r}_k}{(\mathbf{r}_k)^T \mathbf{r}_k}, \tag{150}$$

Repeat

$$\mathbf{z}_{k+1} = \text{prox}_{\alpha_y g}(\mathbf{y}_k - \alpha_y \nabla f(\mathbf{y}_k)), \tag{151}$$

$$\alpha_y = \alpha_y \times \rho, \tag{152}$$

until $F(\mathbf{z}_{k+1}) \leq F(\mathbf{y}_k) - \delta \|\mathbf{z}_{k+1} - \mathbf{y}_k\|^2$ or $F(\mathbf{z}_{k+1}) \leq c_k - \delta \|\mathbf{z}_{k+1} - \mathbf{y}_k\|^2$.
**if** $F(\mathbf{z}_{k+1}) \leq c_k - \delta \|\mathbf{z}_{k+1} - \mathbf{y}_k\|^2$ **then**

$$\mathbf{x}_{k+1} = \mathbf{z}_{k+1}. \tag{153}$$

**else**

$$\mathbf{s}_k = \mathbf{x}_k - \mathbf{y}_{k-1}, \mathbf{r}_k = \nabla f(\mathbf{x}_k) - \nabla f(\mathbf{y}_{k-1}), \tag{154}$$

$$\alpha_x = \frac{(\mathbf{s}_k)^T \mathbf{s}_k}{(\mathbf{s}_k)^T \mathbf{r}_k} \quad or \quad \alpha_x = \frac{(\mathbf{s}_k)^T \mathbf{r}_k}{(\mathbf{r}_k)^T \mathbf{r}_k}, \tag{155}$$

Repeat

$$\mathbf{v}_{k+1} = \text{prox}_{\alpha_x g}(\mathbf{x}_k - \alpha_x \nabla f(\mathbf{x}_k)), \tag{156}$$

$$\alpha_x = \alpha_x \times \rho, \tag{157}$$

until $\quad F(\mathbf{v}_{k+1}) \leq c_k - \delta \|\mathbf{v}_{k+1} - \mathbf{x}_k\|^2$.

$$\mathbf{x}_{k+1} = \begin{cases} \mathbf{z}_{k+1}, & \text{if } F(\mathbf{z}_{k+1}) \leq F(\mathbf{v}_{k+1}), \\ \mathbf{v}_{k+1}, & \text{otherwise.} \end{cases} \tag{158}$$

**end if**

$$t_{k+1} = \frac{\sqrt{4(t_k)^2 + 1} + 1}{2}, \tag{159}$$

$$q_{k+1} = \eta q_k + 1, \tag{160}$$

$$c_{k+1} = \frac{\eta q_k c_k + F(\mathbf{x}_{k+1})}{q_{k+1}}. \tag{161}$$

**end for**

---

where

$$\mathbf{v}_2 = \text{prox}_{\alpha_x g}(\mathbf{x}_1 - \alpha_x \nabla f(\mathbf{x}_1)). \tag{166}$$

If for all $k = 1, \cdots, j$, the conclusions hold, then we consider $k = j + 1$. Define

$$D_{j+1}(t) = \frac{tc_j + F(\mathbf{x}_{j+1})}{t + 1}, \tag{167}$$

then

$$\frac{\mathrm{d}}{\mathrm{d}t} D_{j+1}(t) = \frac{c_j - F(\mathbf{x}_{j+1})}{(t + 1)^2}. \tag{168}$$

If (142) in Algorithm 3 (or (153) in Algorithm 4) is executed, then

$$F(\mathbf{x}_{j+1}) = F(\mathbf{z}_{j+1}) \leq c_j. \tag{169}$$

If (144) in Algorithm 3 (or (158) in Algorithm 4) is executed, by the induction step, we have that $F(\mathbf{v}_{j+1}) \leq c_j - \delta\|\mathbf{v}_{j+1} - \mathbf{x}_j\|$. So

$$F(\mathbf{x}_{j+1}) \leq F(\mathbf{v}_{j+1}) \leq c_j. \tag{170}$$

So we have

$$\frac{\mathrm{d}}{\mathrm{d}t} D_{j+1}(t) \geq 0, \tag{171}$$

which means that $D_{j+1}(t)$ is nondecreasing. So

$$F(\mathbf{x}_{j+1}) = D_{j+1}(0) \leq D_{j+1}(\eta q_j) = c_{j+1}. \tag{172}$$

From the definition of $q_k$ we have

$$q_{k+1} = 1 + \sum_{i=1}^{k} \eta^i < k + 1, \tag{173}$$

due to $\eta \in [0, 1)$. So we have

$$c_{j+1} = D_{j+1}(\eta q_j) = D_{j+1}(q_{j+1} - 1) \tag{174}$$

$$\leq D_{j+1}(j) = \frac{jc_j + F(\mathbf{x}_{j+1})}{j+1} \leq \frac{jA_j + F(\mathbf{x}_{j+1})}{j+1} = A_{j+1}. \tag{175}$$

From (18)-(23) and using $F(\mathbf{x}_{j+1}) \leq c_{j+1}$ we have

$$F(\mathbf{v}_{j+2}) \leq F(\mathbf{x}_{j+1}) - \left(\frac{1}{2\alpha_x} - \frac{L}{2}\right)\|\mathbf{v}_{j+2} - \mathbf{x}_{j+1}\|^2 \tag{176}$$

$$\leq c_{j+1} - \left(\frac{1}{2\alpha_x} - \frac{L}{2}\right)\|\mathbf{v}_{j+2} - \mathbf{x}_{j+1}\|^2. \tag{177}$$

So $\alpha_x < \frac{1}{L}$ such that

$$\mathbf{v}_{j+2} = \mathrm{prox}_{\alpha_x g}(\mathbf{x}_{j+1} - \alpha_x \nabla f(\mathbf{x}_{j+1})) \tag{178}$$

satisfies

$$F(\mathbf{v}_{j+2}) \leq c_{j+1} - \delta\|\mathbf{v}_{j+2} - \mathbf{x}_{j+1}\|^2. \tag{179}$$

$\blacksquare$

**Theorem 4** *Let $f$ be a proper function with Lipschitz continuous gradients and $g$ be proper and lower semicontinuous. Let $\Omega_1 = \{k_1, k_2, \cdots, k_j, \cdots\}$ and $\Omega_2 = \{m_1, m_2, \cdots, m_j, \cdots\}$ such that (142) in Algorithm 3 (or (153) in Algorithm 4) is executed for all $k = k_j \in \Omega_1$ and (144) in Algorithm 3 (or (158) in Algorithm 4) is executed for all $k = m_j \in \Omega_2$. For nonconvex $f$ and nonconvex nonsmooth $g$, assume that (3) holds, then $\{\mathbf{x}_k\}$, $\{\mathbf{v}_k\}$ and $\{\mathbf{y}_{k_j}\}$ where $k_j \in \Omega_1$, generated by Algorithms 3 and 4, are bounded and*

1. *if $\Omega_1$ or $\Omega_2$ is finite, then for any accumulation point $\{\mathbf{x}^*\}$ of $\{\mathbf{x}_k\}$, we have $0 \in \partial F(\mathbf{x}^*)$.*

2. *if $\Omega_1$ and $\Omega_2$ are both infinite, then for any accumulation point $\mathbf{x}^*$ of $\{\mathbf{x}_{k_j+1}\}$, $\mathbf{y}^*$ of $\{\mathbf{y}_{k_j}\}$ where $k_j \in \Omega_1$, and any accumulation point $\mathbf{x}^*$ of $\{\mathbf{x}_{m_j}\}$, $\mathbf{v}^*$ of $\{\mathbf{v}_{m_j+1}\}$ where $m_j \in \Omega_2$, we have $0 \in \partial F(\mathbf{x}^*)$, $0 \in \partial F(\mathbf{y}^*)$ and $0 \in \partial F(\mathbf{v}^*)$.*

**Proof** From Algorithm 3 we know that if (142) (or (153) in Algorithm 4) is executed, then

$$F(\mathbf{x}_{k+1}) \leq c_k - \delta\|\mathbf{x}_{k+1} - \mathbf{y}_k\|^2, \tag{180}$$

and

$$c_{k+1} = \frac{\eta q_k c_k + F(\mathbf{x}_{k+1})}{q_{k+1}} \tag{181}$$

$$\leq \frac{\eta q_k c_k + c_k - \delta\|\mathbf{x}_{k+1} - \mathbf{y}_k\|^2}{q_{k+1}} \tag{182}$$

$$= c_k - \frac{\delta\|\mathbf{x}_{k+1} - \mathbf{y}_k\|^2}{q_{k+1}}. \tag{183}$$

If (144) (or (153) in Algorithm 4) is executed, then

$$F(\mathbf{x}_{k+1}) \leq F(\mathbf{v}_{k+1}) \leq c_k - \delta\|\mathbf{v}_{k+1} - \mathbf{x}_k\|^2, \tag{184}$$

and

$$c_{k+1} \leq c_k - \frac{\delta\|\mathbf{v}_{k+1} - \mathbf{x}_k\|^2}{q_{k+1}}. \tag{185}$$

From $F(\mathbf{x}_{k+1}) \leq c_k \leq A_k = \frac{\sum_{i=1}^{k} F(\mathbf{x}_i)}{k}$ we can have that $F(\mathbf{x}_{k+1})$ and $c_k$ are bounded by induction. By assumption (3) we know that $\{\mathbf{x}_k\}$ is bounded. From $F(\mathbf{v}_{k+1}) \leq c_k$ we know $\mathbf{v}_{k+1}$ is bounded if $\mathbf{v}_{k+1}$ is computed.

From the definitions of $\Omega_1$ and $\Omega_2$, we have

$$c_{k_j+1} \leq c_{k_j} - \frac{\delta\|\mathbf{x}_{k_j+1} - \mathbf{y}_{k_j}\|^2}{q_{k_j+1}}, k_j \in \Omega_1 \tag{186}$$

$$c_{m_j+1} \leq c_{m_j} - \frac{\delta\|\mathbf{v}_{m_j+1} - \mathbf{x}_{m_j}\|^2}{q_{m_j+1}}, m_j \in \Omega_2 \tag{187}$$

$$\Omega_1 \bigcup \Omega_2 = \{1, 2, 3, \cdots\}, \Omega_1 \bigcap \Omega_2 = \emptyset. \tag{188}$$

From the definition of $q_k$ we have

$$q_{k+1} = 1 + \sum_{i=1}^{k} \eta^i = \sum_{i=0}^{k} \eta^i \leq \sum_{i=0}^{\infty} \eta^i = \frac{1}{1-\eta}, \tag{189}$$

So we have

$$\delta(1-\eta)\|\mathbf{x}_{k_j+1} - \mathbf{y}_{k_j}\|^2 \leq \frac{\delta\|\mathbf{x}_{k_j+1} - \mathbf{y}_{k_j}\|^2}{q_{k_j+1}} \leq c_{k_j} - c_{k_j+1}, \tag{190}$$

$$\delta(1-\eta)\|\mathbf{v}_{m_j+1} - \mathbf{x}_{m_j}\|^2 \leq \frac{\delta\|\mathbf{v}_{m_j+1} - \mathbf{x}_{m_j}\|^2}{q_{m_j+1}} \leq c_{m_j} - c_{m_j+1}. \tag{191}$$

where $k_j \in \Omega_1, m_j \in \Omega_2$. Summing over $j = 1, \cdots, \infty$, we have

$$\delta(1-\eta)\sum_{j=1}^{\infty}(\|\mathbf{x}_{k_j+1} - \mathbf{y}_{k_j}\|^2 + \|\mathbf{v}_{m_j+1} - \mathbf{x}_{m_j}\|^2) \leq c_1 - F^*. \tag{192}$$

where $k_j \in \Omega_1, m_j \in \Omega_2$, $F^*$ is the same function value at all the accumulation points and remark that $F(\mathbf{x}_k) \leq c_k$ in Lemma 2, $\Omega_1 \bigcup \Omega_2 = \{1, 2, 3, \cdots\}, \Omega_1 \bigcap \Omega_2 = \emptyset$ and for a fixed $k$, either (183) or (185) holds. So we have

$$\sum_{j=1}^{\infty}(\|\mathbf{x}_{k_j+1} - \mathbf{y}_{k_j}\|^2 + \|\mathbf{v}_{m_j+1} - \mathbf{x}_{m_j}\|^2) \leq \frac{c_1 - F^*}{\delta(1-\eta)} < \infty. \tag{193}$$

We consider three cases one by one.

(1) $\Omega_2$ is finite. In this case, there exists $K_0$ such that (142) (or (153) in Algorithm 4) is executed for all $k > K_0$. So

$$\sum_{k=K_0}^{\infty}\|\mathbf{x}_{k+1} - \mathbf{y}_k\|^2 < \infty, \|\mathbf{x}_{k+1} - \mathbf{y}_k\|^2 \to 0. \tag{194}$$

From the boundness of $\{\mathbf{x}_k\}$ we have that $\{\mathbf{y}_k\}$ is bounded because $\|\mathbf{x}_{k+1} - \mathbf{y}_k\|^2 \to 0$. Let $\mathbf{y}^*$ be any accumulation point of $\{\mathbf{y}_k\}$, say $\{\mathbf{y}_{k_l}\} \to \mathbf{y}^*$ as $l \to \infty$. From $\|\mathbf{x}_{k+1} - \mathbf{y}_k\|^2 \to 0$ we have $\{\mathbf{x}_{k_l+1}\} \to \mathbf{y}^*$ as $l \to \infty$.

From the optimality condition of (141) and $\mathbf{x}_{k+1} = \mathbf{z}_{k+1}$ we have

$$0 \in \nabla f(\mathbf{y}_{k_l}) + \frac{1}{\alpha_y}(\mathbf{x}_{k_l+1} - \mathbf{y}_{k_l}) + \partial g(\mathbf{x}_{k_l+1}) \tag{195}$$

$$= \nabla f(\mathbf{x}_{k_l+1}) + \nabla f(\mathbf{y}_{k_l}) - \nabla f(\mathbf{x}_{k_l+1}) + \frac{1}{\alpha_y}(\mathbf{x}_{k_l+1} - \mathbf{y}_{k_l}) + \partial g(\mathbf{x}_{k_l+1}), \tag{196}$$

So we have

$$-\nabla f(\mathbf{y}_{k_l}) + \nabla f(\mathbf{x}_{k_l+1}) - \frac{1}{\alpha_y}(\mathbf{x}_{k_l+1} - \mathbf{y}_{k_l}) \in \partial F(\mathbf{x}_{k_l+1}), \tag{197}$$

and

$$\left\| \nabla f(\mathbf{y}_{k_l}) - \nabla f(\mathbf{x}_{k_l+1}) + \frac{1}{\alpha_y}(\mathbf{x}_{k_l+1} - \mathbf{y}_{k_l}) \right\| \leq \left( \frac{1}{\alpha_y} + L \right) \|\mathbf{x}_{k_l+1} - \mathbf{y}_{k_l}\| \to 0, \tag{198}$$

as $l \to \infty$.

From (141) and $\mathbf{x}_{k+1} = \mathbf{z}_{k+1}$ we have

$$\langle \nabla f(\mathbf{y}_{k_l}), \mathbf{x}_{k_l+1} - \mathbf{y}_{k_l} \rangle + \frac{1}{2\alpha_y} \|\mathbf{x}_{k_l+1} - \mathbf{y}_{k_l}\|^2 + g(\mathbf{x}_{k_l+1}) \tag{199}$$

$$\leq \quad \langle \nabla f(\mathbf{y}_{k_l}), \mathbf{y}^* - \mathbf{y}_{k_l} \rangle + \frac{1}{2\alpha_y} \|\mathbf{y}^* - \mathbf{y}_{k_l}\|^2 + g(\mathbf{y}^*). \tag{200}$$

So

$$\limsup_{l \to \infty} g(\mathbf{x}_{k_l+1}) \leq g(\mathbf{y}^*). \tag{201}$$

From the definition of lower semicontinuous of $g$ we have

$$\liminf_{l \to \infty} g(\mathbf{x}_{k_l+1}) \geq g(\mathbf{y}^*). \tag{202}$$

So we have

$$\lim_{l \to \infty} g(\mathbf{x}_{k_l+1}) = g(\mathbf{y}^*). \tag{203}$$

Because $f$ is continuously differentiable, we have

$$\lim_{l \to \infty} F(\mathbf{x}_{k_l+1}) = F(\mathbf{y}^*). \tag{204}$$

Similar to Theorem 1 we have

$$0 \in \partial F(\mathbf{y}^*). \tag{205}$$

From $\|\mathbf{x}_{k+1} - \mathbf{y}_k\|^2 \to 0$ we know that $\{\mathbf{x}_k\}$ and $\{\mathbf{y}_k\}$ have the same accumulation point. So for any accumulation point $\mathbf{x}^*$ of $\{\mathbf{x}_k\}$ we have

$$0 \in \partial F(\mathbf{x}^*). \tag{206}$$

(2) $\Omega_1$ is finite. In this case, there exists $K_0$ such that (144) (or (158) in Algorithm 4) is executed for all $k > K_0$. So

$$\sum_{k=K_0}^{\infty} \|\mathbf{v}_{k+1} - \mathbf{x}_k\|^2 < \infty, \|\mathbf{v}_{k+1} - \mathbf{x}_k\|^2 \to 0. \tag{207}$$

Similar to Theorem 1, for any accumulation point $\mathbf{x}^*$ of $\{\mathbf{x}_k\}$ we have

$$0 \in \partial F(\mathbf{x}^*). \tag{208}$$

(3) $\Omega_1$ and $\Omega_2$ are both infinite. In this case

$$\|\mathbf{x}_{k_j+1} - \mathbf{y}_{k_j}\|^2 \to 0, \|\mathbf{v}_{m_j+1} - \mathbf{x}_{m_j}\|^2 \to 0. \tag{209}$$

where $k_j \in \Omega_1, m_j \in \Omega_2$. From the boundness of $\{\mathbf{x}_k\}$ we know $\mathbf{y}_{k_j}$ is bounded where $k_j \in \Omega_1$. From cases 1 and 2, we know that for any accumulation point $\mathbf{y}^*$ of $\{\mathbf{y}_{k_j}\}$, $k_j \in \Omega_1$ and any accumulation point $\mathbf{x}^*$ of $\{\mathbf{x}_{m_j}\}$, $m_j \in \Omega_2$, we have $0 \in \partial F(\mathbf{y}^*)$ and $0 \in \partial F(\mathbf{x}^*)$. Because $\{\mathbf{x}_{k_j+1}\}$ and $\{\mathbf{y}_{k_j}\}$ have the same accumulation point for $k_j \in \Omega_1$, $\{\mathbf{v}_{m_j+1}\}$ and $\{\mathbf{x}_{m_j}\}$ have the same accumulation point for $m_j \in \Omega_2$. So for any accumulation point $\mathbf{x}^*$ of $\{\mathbf{x}_{k_j+1}\}$, $k_j \in \Omega_1$, and any accumulation point $\mathbf{v}^*$ of $\{\mathbf{v}_{m_j+1}\}$, $m_j \in \Omega_2, 0 \in \partial F(\mathbf{x}^*), 0 \in \partial F(\mathbf{v}^*)$. ∎

Table 1: Comparisons of APG, PG, GPower and CurveLS on the Sparse PCA problem. The quantities include number of iterations, computing time (in seconds), sparsity (percentage of zeros) and adjusted variance. We pursuit high sparsity and variance. They are averaged over 10 runs.

| m | Method | #Iter. | Time | sparsity | var |
|---|--------|--------|------|----------|-----|
| 40 | GPower | 1557 | 697 | 0.5341 | 0.5532 |
| | PG | 1554 | 695 | 0.5341 | 0.5532 |
| | CurviLS | 647 | 318 | 0.5343 | 0.5541 |
| | mAPG | 275 | 268 | 0.5342 | 0.5536 |
| | nmAPG | 385 | 202 | 0.5341 | 0.5539 |
| 60 | GPower | 1315 | 711 | 0.5992 | 0.6048 |
| | PG | 1316 | 716 | 0.5992 | 0.6048 |
| | CurviLS | 790 | 474 | 0.5991 | 0.6047 |
| | mAPG | 268 | 322 | 0.5994 | 0.6049 |
| | nmAPG | 364 | 225 | 0.5994 | 0.6049 |
| 80 | GPower | 1574 | 1012 | 0.6457 | 0.6367 |
| | PG | 1575 | 1009 | 0.6457 | 0.6367 |
| | CurviLS | 941 | 662 | 0.6455 | 0.6366 |
| | mAPG | 262 | 371 | 0.6457 | 0.6370 |
| | nmAPG | 391 | 282 | 0.6459 | 0.6373 |

## 4 Numerical Results: Sparse PCA

Principal Component Analysis (PCA) is a basic technique for finding low-dimensional representations. But it has a drawback of lack of interpretability. Sparse PCA is a common approach to find interpretable principal components and has been applied successfully in areas such as bioinformatics [7]. One of the most popular approaches for solving Sparse PCA is the Generalized Power Method (GPower) [8]. It first solves the following problem (210), then adds a post-processing step. We focus here on the time consuming problem (210), which is an optimization problem on the Stiefel manifold:

$$\min_{X^T X=I} f(X) = -\frac{1}{2} \sum_{j=1}^{m} \sum_{i=1}^{d} [\mu_j |\mathbf{a}_i^T \mathbf{x}_j| - \gamma_j]_+^2, \tag{210}$$

where $X \in \mathbb{R}^{n \times m}$, $n$ is the sample size, $m$ is the desired number of PCA component, $A \in \mathbb{R}^{n \times d}$ is the data matrix, $d$ is the sample dimension and $\gamma$ controls the sparsity. $[x]_+ = \max\{x, 0\}$. We set $\mu_j = 1$, $\gamma_j = 0.2$ for all $1 \le j \le m$, and test with different $m$'s.

We compare monotone APG (mAPG) and nonmonotone APG (nmAPG) with Proximal Gradient Method (PG), GPower and the Curvilinear search method (CurviLS) [9], the state-of-art algorithm on the Stiefel manifold. The performance of PG and Inertial Forward-Backward (IFB) is similar. So we omit to list the result of IFB here. We test the performance on the breast cancer data set[1], which contains 295 samples of 8241 dimensions. All the algorithms are terminated when $\|Df(X)\|_\infty < 0.1$ or the number of iterations exceeds 3000, where $Df(X) := \nabla f(X) - X(\nabla f(X))^T X$ is the projected gradient onto the tangent planes. We test the machine learning performance by the sparsity and the adjusted variance [10]. In PG and APG, we set the stepsize $\alpha = 100$. $f(X)$ in (210) is a concave function and any stepsize can ensure that $F(\mathbf{v}_{k+1}) \le F(\mathbf{x}_k) - \delta\|\mathbf{v}_{k+1} - \mathbf{x}_k\|^2$ holds. So we choose a large stepsize to make it close to that of GPower, which can be viewed as using a stepsize of $\infty$.

Table 1 shows the related result. We can note that APG-type algorithms are much faster than PG and GPower. mAPG needs fewer iterations while nmAPG needs less time. On the one hand, this indicates that the monitor-corrector step in mAPG takes effect. On the other hand, the cost of each iteration in mAPG is almost twice than that of nmAPG. This means that in nmAPG $F(\mathbf{z}_{k+1}) \le c_k - \delta\|\mathbf{z}_{k+1} - \mathbf{y}_k\|^2$ holds almost in all iterations and accordingly $\mathbf{v}_k$ is not computed in most of the time. We can also see that APG-type algorithms are faster than CurviLS, demonstrating that APG is a competitive method for optimization on the Stiefel manifold.

## Footnotes

[1]Data available at http://cbio.ensmp.fr/ ljacob/documents/overlasso-package.tgz