[Reviews · NeurIPS 2015]

Submitted by Assigned_Reviewer_1

This paper extends the famous accelerated proximal gradient (APG) methods to nonconvex programming. The novelty lies in a new auxiliary sequence that guards the quality of the extrapolation sequence. The authors show that all the accumulation points generated by the proposed algorithms are critical points and the convergence rate remains 1/k^2 for convex problems. However, the coercive assumption may weaken the contribution of this paper.

Quality: The authors study a very important topic in the communities of optimization and machine learning. By a newly-invented device (the monitor sequence), they extend many desirable properties of the APG algorithms for the convex problems to the nonconvex problems. This paper is very technical and I found the proofs are rigorous. However, my major concern in the technical part is that the coercive assumption may be too strong and exclude many important models in machine learning, e.g., logistic and least square loss with p >> n design matrix and l_0 penality. Moreover, experiments (Table 2) show that the proposed non-monotone APG leads to the highest error.

Clarity: Many important details are missing in the main text. I think the main text can be much clearer if the authors move some materials from the supplement to the main text, e.g., the detailed and complete description of Algorithms 1 and 3. The presentation in the supplement is quite clear.

Originality: The major novelty lies in a new auxiliary sequence that guards the quality of the extrapolation sequence.

Significance: As mentioned by the authors, this paper is the first to generalize APG type algorithms to general nonconvex and nonsmooth problems with some desirable properties.

However, my major concern in the technical part is that the coercive assumption may be too strong and exclude many important models in machine learning, e.g., logistic and least square loss with p >> n design matrix and l_0 penality. This may weaken the contribution of this paper.

Summary: This paper extends the famous accelerated proximal gradient (APG) methods to nonconvex programming. The novelty lies in a new auxiliary sequence that guards the quality of the extrapolation sequence. The authors show that all the accumulation points generated by the proposed algorithms are critical points and the convergence rate remains 1/k^2 for convex problems. However, the coercive assumption may weaken the contribution of this paper.

Submitted by Assigned_Reviewer_2

The paper proposes an accelerated gradient-based algorithm for

non-convex composite optimization. Essentially, it provides an

alternative to the existing work of Ghadimi and Lan, 2013. The main

features of the algorithm is to preserve Nesterov's acceleration for

convex problems, while providing convergence to stationary points for

non-convex ones. The paper also presents convergence analysis from

various points of views.

I like the paper and believe that it is a strong contribution.

Nevertheless, there are a few points that could improve the paper

- I found the comparison with Ghadimi and Lan not quite fair. On the one

hand, their approach requires a convex function g and does not apply to

the more general problem considered in the paper. On the other hand,

their convergence results are significantly stronger than simple convergence

to stationary points, since they also provide convergence rates.

- The experimental part is poor compared to the rest of the paper. The purpose

of this experiments should be to convince the reader that acceleration

applies in practice. To settle the case, it would have been good to

simply report the objective function achieved by the different methods as

as function of the number of iterations. Instead, the table reports a

comparison for a specific arbitrary objective function value. Why

this objective function value makes sense is not clear. It is also

not clear that conclusions drawn for this value would generalize to

other ones.

- Finally, I found the experimental setup presented in the

supplementary material (sparse PCA) more interesting than the one

presented in the main paper. sparse PCA is more challenging, and more

interesting. This is of course subjective, but perhaps, the authors

should consider including the sparse PCA experiment in the main paper

instead of the logistic regression one.

Misc comments:

- Please define and comment on the KL assumption in the final version

of the paper, if it is accepted. The paper should be self-content

without having to read the supplementary material.

Summary: This is a solid contribution, well presented, discussed, and

analyzed. Only the experimental part is poor.

Submitted by Assigned_Reviewer_3

The paper presents accelerated proximal gradient methods for nonconvex and nonsmooth problems by introducing a monitor satisfying the sufficient descent property to existing APG methods devised for convex programs. This is the first work which provides APG-type algorithms for general nonconvex and nonsmooth problems ensuring that every accumulation point is a critical point. The paper also proves that the convergence rates remain O(1/k^2) when the problems are convex.

The paper is very well-written. The proposed APG-type algorithm for nonconvex problems is of great significance because every accumulation point is proved to be a critical point and the convergence rate is also investigated. The paper will be more easy to read if the following concerns are addressed properly.

1: Supplementary material shows the definition of KL property, which implies the role of the desingularising function \varphi(t). However, the main paper does not discuss the desingularising function \varphi(t) and suddenly \varphi(t) appears in Theorem 3 (seemingly unrelated to the KL property). Though it may be difficult to find a space to write the KL property using \varphi(t), can the authors briefly discuss how to use the desingularising function \varphi(t) in KL property?

2: Can the authors show the examples of nonconvex problems satisfying all conditions on f and g shown in Theorem 3?

3: I understood that the convergence rate of nonmonotone APG has the same convergence rate to that of mononote APG in the convex case. Can the authors compare the computational complexities of Algorithm 1 and Algorithm 2 in one iteration?
Summary: The paper is very well-written and the APG-type algorithm with theoretical guarantee for nonconvex problems is of great significance.

Author Feedback
Author rebuttal: We thank all the reviewers for being positive towards our paper. We will carefully address their comments. Below are some clarifications:

R#1:

1.The coercive assumption may be too strong.

For all we know, the coercive assumption is a standard assumption in nonconvex analysis of first order descent algorithms, e.g., proximal gradient method and its variants, see references [18,19,20,21,22,23]. It is used to make sure that the sequence generated by the algorithm is bounded, thus the accumulation point exists. It may be a difficult job for nonconvex analysis of first order algorithm if the coercive assumption is absent. However, we will consider this problem in our further work.
When solving logistic and least square loss problems with p >> n design matrix and l_0 penalty, the coercive assumption can be avoided if we have the prior knowledge that |x_i|<=C for all 1<=i<=p, as this can also imply that the iterations are bounded. We may incorporate this box constraint into the optimization model and the subproblem at each step is still easily solvable due to the particularity of l_0 penalty.

2.The proposed non-monotone APG leads to the highest error.

The proposed monotone APG also leads to the lowest error. The difference is tiny (0.04%). We think this is because different algorithms convergence to different critical points in nonconvex programs and thus make the subsequent machine learning performance a little different.

R#2:

1.The comparison with Ghadimi and Lanis not quite fair.

In the final version, we will mention more about their contributions on convergence rate.

2.The experimental part is poor.

We have done the required experiments. They show that the objective function values of our method w.r.t. iteration number are always lower than those of other methods. Similar phenomenon can be observed on objective value vs. CPU time. We will add the curves in the revised paper.

R#3
1.Discuss how to use the desingularising function \varphi(t) in KL property.

In KL property, the desingularising function \varphi(t) plays the role of sharping f when approaching to the critical points. Indeed, the KL inequality can be proved to imply dist(0, \partial \varphi(f(u)-f(u^*)))>=1, where u^* is the critical point. This means that the subgradient of function \varphi(f(u)-f(u^*)) have a norm greater than 1, no matter how close is u to u^* [23].

2.Show the examples of nonconvex problems satisfying all conditions on f and g shown in Theorem 3.

For example, f can be real polynomial functions. g can be \|x\|_p where p>=0, rank(X), the indicator function of PSD cone, Stiefel manifolds and constant rank matrices [23].

3.Compare the computational complexities of Algorithm 1 and Algorithm 2 in one iteration.

When line search is not used, Alg 1 needs 2 gradient computations, 2 proximal mappings and 2 function evaluations at each step. Alg 2 has a "if else" choice. If the "if" branch is chosen, then Alg 2 needs 1 gradient computation, 1 proximal mapping and 1 function evaluation in one iteration. If the "else" branch is chosen, then 2 gradient computations, 2 proximal mappings and 2 function evaluations are needed. In the sparse PCA experiment, Alg 2 runs 354 iterations and 335 of which choose the "if" branch. Thus in average Alg 2 needs 1.05 gradient computations, 1.05 proximal mappings and 1.05 function evaluations at each step.
When line search is used, let r_1 and r_2 be the number of line searches for z and v. Alg 1 needs 3 gradient computations, r_1+r_2 proximal mappings and r_1+r_2+1 function evaluations at each step. If the "if" branch in Alg 2 is chosen, then Alg 2 needs 1 gradient computation, r_1 proximal mappings and r_1 function evaluations at each step. If the "else" branch is chosen, then 2 gradient computations, r_1+r_2 proximal mappings and r_1+r_2+1 function evaluations are needed. In the sparse logistic regression experiment, Alg 1 needs 3 gradient computations, 2.20 proximal mappings and 3.20 function evaluations in average for one running. Alg 2 needs 1.007 gradient computations, 1.007 proximal mappings and 1.014 function evaluations.

We will incorporate all the above clarifications in the revised paper.

We will also pay attention to other suggestions, for instance, move some materials from the supplement to the main text, define and comment on the KL assumption and include the sparse PCA experiment in the main paper instead of the logistic regression one.